# LC3B is an RNA-binding protein to trigger rapid mRNA degradation during autophagy

Hyun Jung Hwang [1,2,3], Hongseok Ha [1,2,3], Ban Seok Lee[1,2], Bong Heon Kim[2], Hyun Kyu Song [2] &
Yoon Ki Kim [1,2 ✉]

LC3/ATG8 has long been appreciated to play a central role in autophagy, by which a variety of cytoplasmic materials are delivered to lysosomes and eventually degraded. However, information on the molecular functions of LC3 in RNA biology is very limited. Here, we show that LC3B is an RNA-binding protein that directly binds to mRNAs with a preference for a consensus AAUAAA motif corresponding to a polyadenylation sequence. Autophagic activation promotes an association between LC3B and target mRNAs and triggers rapid degradation of target mRNAs in a CCR4-NOT–dependent manner before autolysosome formation. Furthermore, our transcriptome-wide analysis reveals that *PRMT1* mRNA, which encodes a negative regulator of autophagy, is one of the major substrates. Rapid degradation of *PRMT1* mRNA by LC3B facilitates autophagy. Collectively, we demonstrate that LC3B acts as an RNA-binding protein and an mRNA decay factor necessary for efficient autophagy.

[1] Creative Research Initiatives Center for Molecular Biology of Translation, Korea University, Seoul 02841, Republic of Korea. [2] Division of Life Sciences, Korea University, Seoul 02841, Republic of Korea. [3] These authors contributed equally: Hyun Jung Hwang, Hongseok Ha. ✉email: yk-kim@korea.ac.kr

Autophagy is a self-eating, intracellular degradation pathway that removes various cellular materials, including unnecessary or dysfunctional components, toxic protein aggregates, damaged organelles, and intracellular microbes, using lysosomes in either a selective or nonselective manner[1–4]. Autophagy is minimally activated under normal conditions and is dynamically activated by a broad range of cellular stressors, such as nutrient deprivation, consequently functioning in self-nourishment or cellular homeostasis. Accordingly, it is known that autophagy is linked to diverse biological or physiological events, including osteoarthritis, tumorigenesis, inflammation, and neurodegenerative disorders[2,4,5].

Depending on the modes of recognition and elimination of substrates, autophagy is divided into three types: chaperone-mediated autophagy, microautophagy, and macroautophagy (the best-characterized type of autophagy, hereafter referred to as autophagy). Autophagy is initiated by sequestering cytoplasmic materials into double-membrane autophagosomes[6]. After that, autophagosomes fuse with lysosomes to form autolysosomes. The sequestered materials are eventually degraded within autolysosomes. All these steps are tightly regulated by evolutionarily conserved autophagy-related (ATG) proteins, such as ATG8, the key autophagic ubiquitin-like protein involved in the formation of autophagophores[7,8]. Mammalian ATG8 comprises three subfamilies: microtubule-associated protein 1 light chain 3 (MAP1LC3, hereafter referred to as LC3), Golgi-associated ATPase enhancer of 16 kDa (GATE-16), and gamma-aminobutyric acid receptor-associated protein. Mammalian LC3 isoforms (LC3A, LC3B, and LC3C) are subject to a posttranslational ubiquitin-like conjugation pathway, by which the LC3s are covalently conjugated to phosphatidylethanolamine (PE) and thereby selectively localize to the autophagosomal membrane[9,10]. This conversion of LC3 from an unconjugated form (LC3-I) to a PE-conjugated form (LC3-II) is pivotal for the formation of autophagosomes[5].

In eukaryotes, mRNAs are present as complexes with a variety of RNA-binding proteins (RBPs), forming messenger ribonucleoprotein (mRNP) complexes[11–14]. The mRNPs are subject to dynamic remodeling during their life with changes in the composition of the RBPs, secondary or tertiary structure, posttranslational modifications of the RBPs, and RNA sequence modifications. The associated RBPs dictate the fate of mRNAs, including export from the nucleus to the cytoplasm, intracellular localization, translation efficiency in the cytoplasm, and stability of mRNAs. In accordance with the regulation of gene expression in multilayers by RBPs, several recent reports have highlighted the interplay between RBPs and autophagy[15–17]. For instance, some RBPs influence alternative splicing, stability, or translation of a subset of *ATG* mRNAs, consequently affecting autophagy[18–21]. In addition, direct binding of a vault RNA (a small non-coding RNA) to p62 (a well-known autophagy receptor) interferes with p62 multimerization, controlling autophagy[22].

In this study, we demonstrate that LC3B, a central protein in autophagy, is an RBP with a binding preference for the AAUAAA consensus motif. Direct binding of LC3B to mRNAs elicits rapid degradation of target mRNAs. We also observe that the LC3B-mediated rapid decay of *PRMT1* mRNA promotes efficient autophagy. Our observations uncover a previously unappreciated role of LC3B as both an RBP and mRNA decay factor in efficient autophagy, highlighting the biological impact of RBP-mediated gene regulation on autophagy.

## Results

**LC3B is an RNA-binding protein.** Several studies have provided a molecular implication that LC3 may have an RNA-binding ability[23–25], although LC3 is not listed as an RBP in recent proteome-wide screening for RBPs in human cells[12,13,26,27]. To investigate the RNA-binding ability of LC3 at the transcriptome level, we performed cross-linking immunoprecipitation (IP) coupled with high-throughput sequencing (CLIP-seq) with an antibody against endogenous LC3B, the best-characterized isoform of human LC3 (Fig. 1a–d and Supplementary Fig. 1). This approach mainly allows us to monitor the direct interaction between RNA and protein within cells[28]. To enrich putative LC3B-bound transcripts and minimize a possible change in the abundance of transcripts via lysosome-mediated RNA degradation, CLIP-seq experiments were performed on HEK293T cells either treated or not treated with both rapamycin (Rapa, which inhibits mTOR kinase activity and induces autophagy) and chloroquine (CQ, which blocks the fusion of autophagosomes with lysosomes, consequently inhibiting a late step of autophagy). Under the conditions of the cells treated with both Rapa and CQ (Rapa + CQ), we observed efficient conversion of LC3B-I (an unconjugated form) to LC3B-II [a PE-conjugated form; Supplementary Fig. 1a], which is pivotal for the formation of autophagosomes[5,9,10].

Raw reads obtained from two biological replicates of CLIP-seq experiments were processed as described in the "Methods" section (Supplementary Data 1a, b). After peak calling and filtering with fragments per kilobase of transcript per million mapped reads (FPKM) ≥ 1, we obtained 1870 common peaks covering 1508 mRNAs in both CLIP1 and CLIP2 of LC3B in the untreated cells and 748 common peaks covering 682 mRNAs in the cells treated with Rapa + CQ. Among them, ~95% and ~99% of LC3B-bound mRNAs showed one or two LC3B peaks in cells either untreated or treated with Rapa + CQ, respectively (Fig. 1a and Supplementary Data 2). Similarly, ~90% and ~96% of LC3B-bound transcripts harbored one or two LC3B peaks in the cells either untreated or treated with Rapa + CQ, respectively (Supplementary Fig. 1b). Notably, the metagene analysis of mRNAs harboring LC3B peaks showed significant enrichment of LC3B peaks in the distal end of the 3′ untranslated region (3′UTR), regardless of the treatment with Rapa + CQ (Fig. 1b). Consensus motif analysis of the peaks present throughout mRNAs (Supplementary Fig. 1c) or located at the 3′UTR (Fig. 1c) using MEME or HOMER revealed "AAUAAA" as a consensus motif for LC3B binding with high significance. However, we could not find any significant consensus motif present in mRNAs harboring LC3B peaks at the 5′UTR or CDS. The identified AAUAAA consensus motif almost completely overlapped with a previously well-characterized polyadenylation signal (PAS; Fig. 1d).

To corroborate the preferential binding of LC3B to the AAUAAA motif, we performed an electrophoretic mobility shift assay (EMSA) using Cy5-labeled triple repeats of AAUAAA or AAAAAA and purified recombinant LC3B (a cleaved form of proLC3B by ATG4). As negative controls, we used bovine serum albumin (BSA) and an LC3B mutant (LC3B-R/Q), which harbors three amino acid substitutions from arginine to glutamine in arginine-rich motifs (amino acids 68, 69, and 70 residues; Supplementary Fig. 2a). We chose these residues within LC3B because the arginine-rich motif is one of the common features found in many RBPs[29–31], located at the surface of LC3B (Supplementary Fig. 2b), and is highly conserved among different species (Supplementary Fig. 2c). Furthermore, the arginine-rich motif has been implicated to be involved in association between LC3B and fibronectin mRNA[23–25]. We observed a distinct shift of the probe relative to Cy5-AAAAAAx3 when LC3B was mixed with Cy5-AAUAAAx3 (Fig. 1e). In contrast, the addition of either BSA or LC3B-R/Q failed to cause a shift in Cy5-AAUAAAx3 (Supplementary Fig. 2d). Preferential interaction between LC3B and AAUAAA motif was further evident by fluorescence polarization assay using purified LC3B-WT or LC3B-R/Q (Fig. 1f). Collectively, our CLIP-seq data and EMSA data indicate

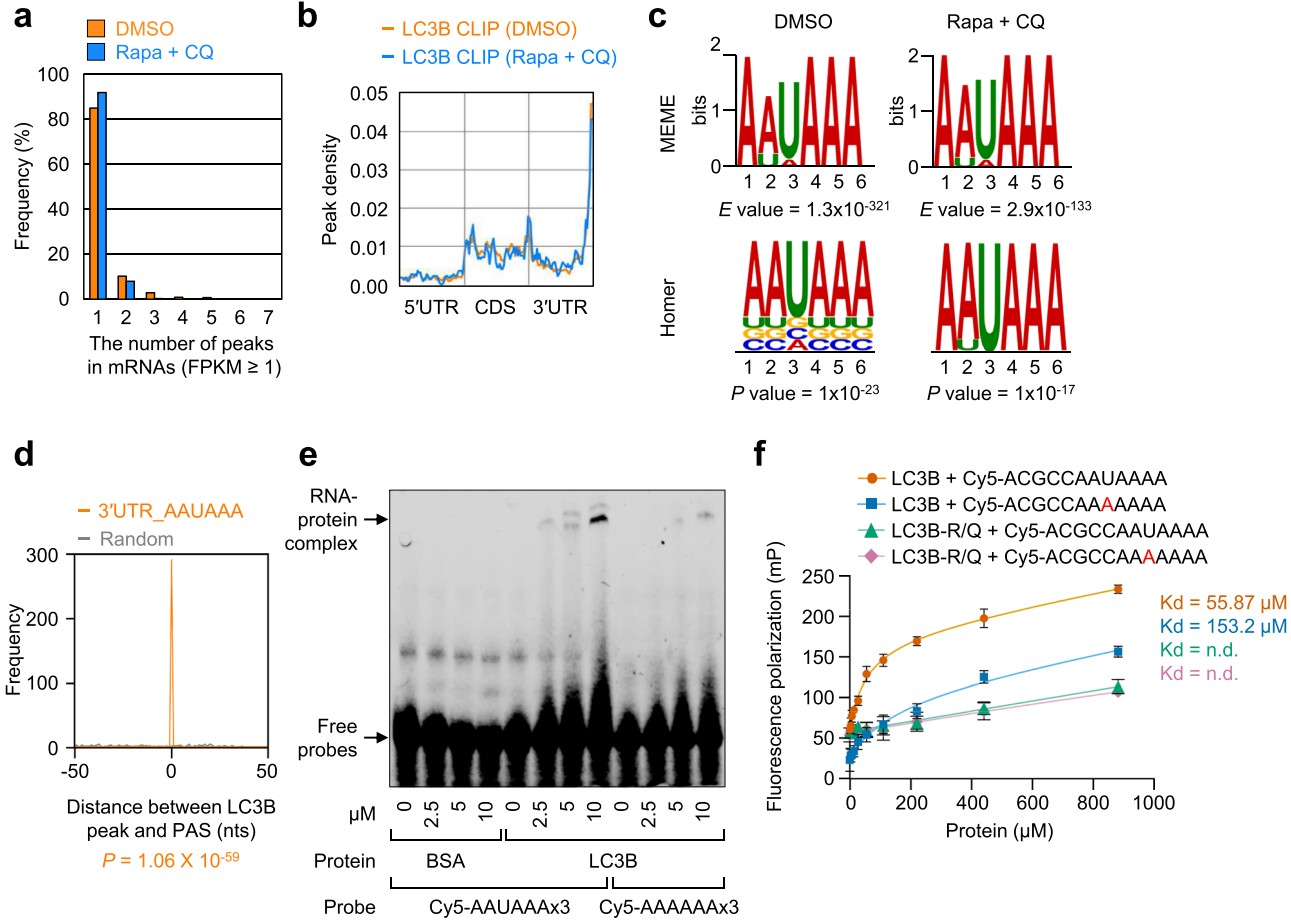

**Fig. 1 LC3B is an RNA-binding protein with a preference for AAUAAA consensus motif. a–d** CLIP-seq of endogenous LC3B on HEK293T cells treated with either DMSO or Rapa + CQ. Data obtained from two biological replicates of LC3B CLIP-seq were analyzed. The mRNAs with FPKM ≥ 1 in the DMSO-treated cells were used for the calculations. **a** The number of LC3B CLIP peaks in mRNAs. **b** Metagene profiles of LC3B peaks. Each region of the 5′UTR, CDS, or 3′UTR was binned into 50 segments. **c** Consensus motif for LC3B binding. The consensus RNA sequences from LC3B peaks located in the 3′UTR in the cells treated with either DMSO (left) or Rapa + CQ (right) were predicted by MEME (upper) or HOMER (lower). The *E* value (upper) estimates the expected number of motifs with a similarly sized set of random sequences using log likelihood ratio. The *p* values (lower) were calculated using cumulative binomial distributions. **d** Profiles of the distance between LC3B peaks located at the consensus AAUAAA motif and polyadenylation signal (PAS). *p* values were calculated using the two-tailed Kolmogorov–Smirnov test. **e** EMSA analysis using in vitro-synthesized Cy5-labeled triple repeats (×3) of either AAUAAA or AAAAAA and either purified recombinant LC3B-WT or BSA. The relative positions of the free probes and RNA-protein complexes are indicated using arrows. Representative data obtained from two independently performed biological replicates (*n* = 2) are shown. **f** Fluorescence polarization assay showing preferential interaction between purified recombinant LC3B and AAUAAA motif. Dissociation constants (Kd) are indicated at the right side of graph; *n* = 6; Data are presented as mean values ± SD; nd not determined. Source data are provided as a Source Data file.

that LC3B directly binds to RNA with a preference for the AAUAAA motif.

**LC3B elicits rapid mRNA degradation in response to autophagic activation.** What happens when LC3B binds to mRNA? To address this question, we assessed the relative changes in the abundance of LC3B-bound mRNAs relative to LC3B-unbound mRNAs at the transcriptome level when the cells were treated with Rapa (Fig. 2a–c; Supplementary Fig. 3a–c; Supplementary Data 1c). A cumulative distribution function (CDF) analysis showed that the relative amounts of mRNAs harboring LC3B peaks (LC3B CLIP group) were significantly reduced upon Rapa treatment, compared with those of nontarget mRNAs (non-targets group; Fig. 2a). Notably, the amounts of mRNAs were more significantly reduced when LC3B was bound to the 3′UTR, relative to the 5′UTR or coding sequence (CDS; Fig. 2b). This reduction was reversed when endogenous LC3B was down-regulated (Fig. 2c).

The mRNAs harboring LC3B peaks in the 3′UTR were further categorized into two groups: 3′UTR_AAUAAA group (which harbors LC3B peaks at the consensus AAUAAA motif) and 3′UTR_No_AAUAAA group (which harbors LC3B peaks elsewhere rather than AAUAAA). Although both groups showed similar reads per FPKM (Supplementary Fig. 3b) and a significant reduction in abundance upon Rapa treatment compared with non-targets (Supplementary Fig. 3c), 3′UTR_AAUAAA was more significantly reduced than 3′UTR_No_AAUAAA (Supplementary Fig. 3c), indicating a preferential reduction of mRNAs harboring the LC3B peak at the AAUAAA motif in the 3′UTR. The observed reduction in the 3′UTR_AAUAAA group was markedly reversed when endogenous LC3B was downregulated (Supplementary Fig. 3d and Supplementary Data 2). Notably, the selective reduction of mRNAs harboring the LC3B peaks at the 3′UTR and its recovery after LC3B downregulation did not significantly correlate with the distance between the translation termination codon and LC3B peak position (Supplementary Fig. 4a, b and Supplementary Data 2). In addition, alternative

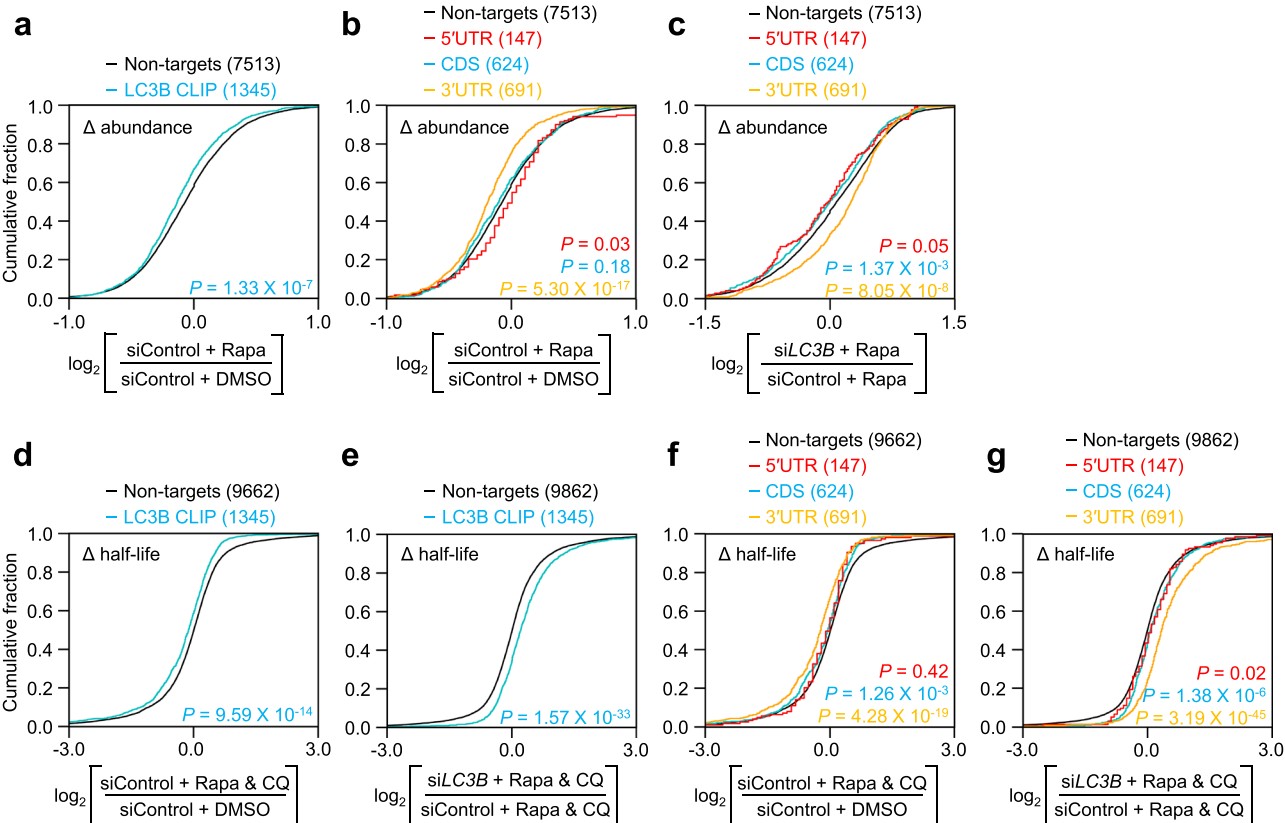

**Fig. 2 Binding of LC3B to target mRNAs triggers rapid mRNA degradation. a–c** Profiles of mRNA abundance. HEK293T cells either undepleted or depleted of endogenous LC3B were treated with either DMSO or Rapa. Total cell RNAs were purified and subjected to mRNA sequencing. The mRNAs with FPKM ≥ 1 in the cells treated with Control siRNA and DMSO were used for the calculations. **a** Cumulative distribution function (CDF) plots for relative changes in mRNA abundance after Rapa treatment. The mRNAs were categorized into two groups: total mRNAs either harboring (LC3B CLIP group) or lacking (non-targets group) the LC3B CLIP peak. CDF plots for the relative changes in the abundance of mRNAs upon Rapa treatment in the undepleted cells (**b**) or upon LC3B downregulation in Rapa-treated cells (**c**). The mRNAs belonging to the LC3B CLIP group were categorized into three groups: 5′UTR, CDS, and 3′UTR depending on the position of the LC3B peak. **d–g** Profiles of mRNA half-life. HEK293T cells either undepleted or depleted of LC3B were treated with either DMSO or Rapa + CQ. The cells were harvested at three time points (0, 6, and 12 h) and total cell RNAs were subjected to mRNA sequencing experiments as described in the "Methods" section. CDF plots for the relative change in the half-life of mRNAs harboring (LC3B CLIP group) or lacking (non-targets group) LC3B peak upon Rapa + CQ treatment in the undepleted cells (**d**) or upon LC3B downregulation in the cells treated with Rapa + CQ (**e**). CDF plots for the relative change in the half-life of mRNAs belonging to the 5′UTR, CDS, or 3′UTR group upon Rapa + CQ treatment in the undepleted cells (**f**) or upon LC3B downregulation in the cells treated with Rapa + CQ (**g**). p values were calculated using the two-tailed Mann–Whitney U test. All NGS data were obtained from two independently performed biological replicates (n = 2). Source data are provided as a Source Data file.

polyadenylation was only marginally affected by Rapa treatment and LC3B downregulation (Supplementary Fig. 4c, d).

We further analyzed the changes in half-life at the transcriptome level (Fig. 2d–g; Supplementary Fig. 3e, f; Supplementary Data 1d). The half-life of LC3B CLIP mRNAs was significantly reduced upon Rapa and CQ treatment compared to that of the nontarget mRNAs (Fig. 2d). In addition, the downregulation of LC3B caused a preferential increase in the half-life of LC3B CLIP mRNAs. (Fig. 2e). In agreement with the change in abundance (Fig. 2b, c), we also observed a preferential reduction in the half-life of mRNAs harboring LC3B peaks at 3′UTR (3′UTR group; Fig. 2f) and 3′UTR_AAUAAA groups (Supplementary Fig. 3e). The observed reduction in half-life was reversed by LC3B downregulation (Fig. 2g and Supplementary Fig. 3f). These data indicate that LC3B-bound mRNAs were rapidly destabilized in response to autophagic activation, which is hereafter referred to as LC3B-mediated mRNA decay (LMD).

**PRMT1 mRNA is targeted for LC3B-mediated mRNA decay.** Top-ranked LMD substrates obtained from our transcriptome analysis (Supplementary Data 2) were validated by quantitative real-time polymerase chain reaction (qRT-PCR). All the tested LMD-targeted mRNAs were efficiently downregulated upon treatment of the cells with either Rapa or Rapa + CQ (Supplementary Fig. 5). This reduction was reversed when the cells were depleted of endogenous LC3B, ATG5, or ATG7 (the latter two are essential factors for the conversion of LC3B-I to LC3B-II[9,10]), suggesting that efficient LMD requires the conversion of LC3B-I to LC3B-II. Among the tested LMD substrates, PRMT1 protein has been previously characterized as a negative regulator of autophagy[32]. Therefore, we investigated the molecular properties of LMD using PRMT1 reporter mRNAs.

PRMT1 mRNA had a single LC3B peak at the 3′-terminal region of the 3′UTR (Fig. 3a). The entire 3′UTR of PRMT1 mRNA was inserted into the Renilla luciferase (RLuc) reporter mRNA (RLuc-P3′-WT; Fig. 3b). As a negative control, the entire 3′UTR of PRMT1 mRNA harboring a single nucleotide substitution (AAUAAA to AAAAAA) was also generated (RLuc-P3′-U/A). Consistent with our EMSA data (Fig. 1e, f), in vivo CLIP experiments, which allowed for mainly monitoring of a direct interaction between transcripts and proteins within cells, revealed

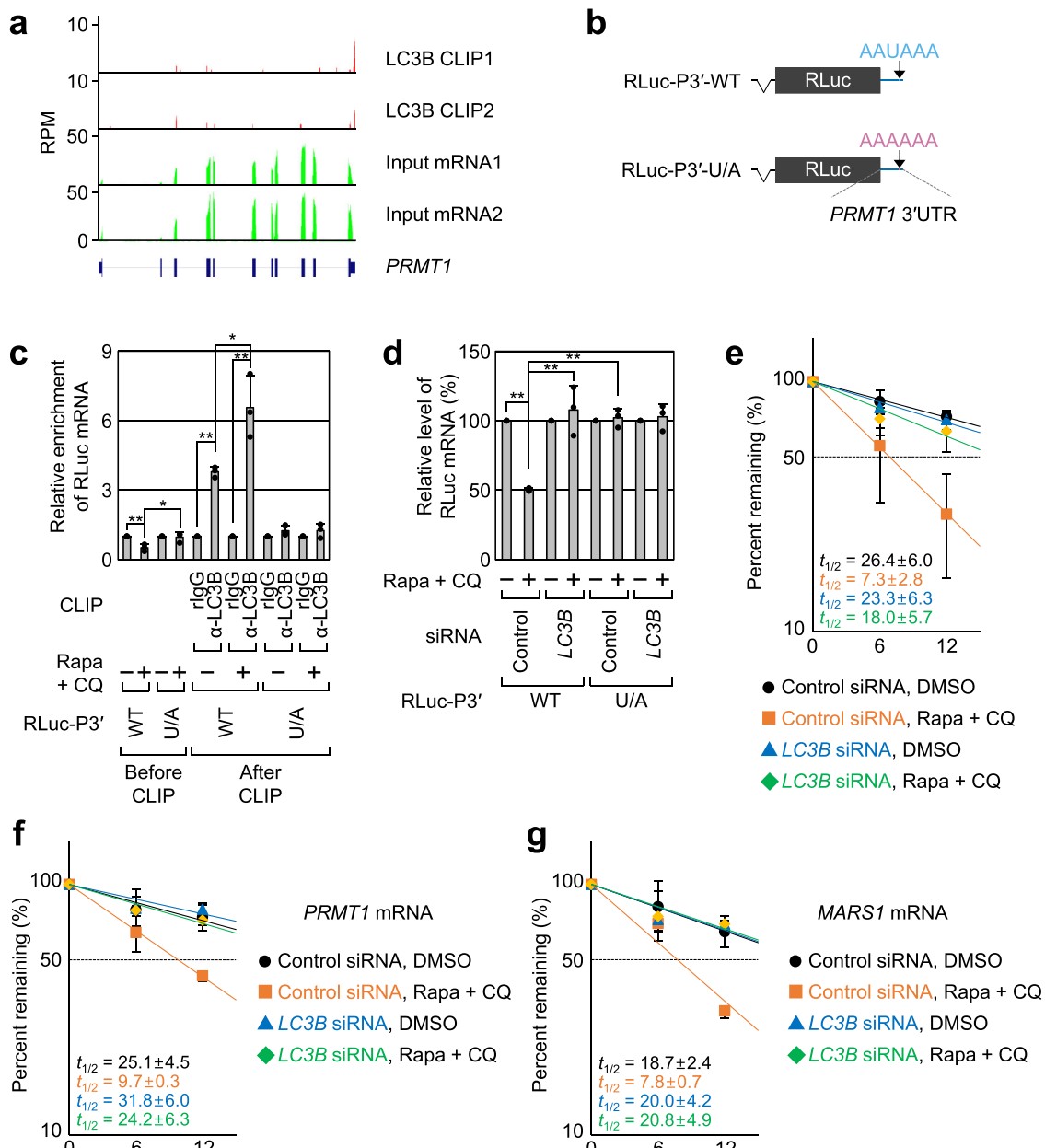

**Fig. 3 *PRMT1* mRNA is a de novo target of LC3B-mediated mRNA decay (LMD). a** Read densities of the LC3B CLIP-seq (LC3B CLIP1 and LC3B CLIP2) and mRNA sequencing (input mRNA1 and input mRNA2) for *PRMT1* mRNA in two biological replicates. The map for *PRMT1* is shown at the bottom. **b** Schematic representation of RLuc reporter mRNAs harboring the full-length *PRMT1* 3′UTR of either WT or U/A substitution. The relative positions of the LC3-binding site (AAUAAA) and its variant (AAAAAA) are indicated by arrows. **c** In vivo CLIP of endogenous LC3B. HEK293T cells expressing the RLuc reporter mRNA and FLuc mRNA (which served as a negative control) were either treated or not treated with Rapa + CQ. The cells were subjected to in vivo CLIP using α-LC3B antibody or nonspecific rabbit IgG (rIgG). The amounts of coimmunoprecipitated reporter mRNAs were normalized to those of the FLuc mRNAs. Then, the normalized levels obtained in IPs using rIgG in the untreated cells were arbitrarily set to 1.0. *n* = 3; Data are presented as mean values ± SD; *p* values were analyzed using two-tailed and equal-variance Student's *t* test; *\*p* < 0.05; *\*\*p* < 0.01 (The exact *p* values are provided in Source Data file). **d** Effect of LC3B downregulation on the abundance of the RLuc-P3′ reporter mRNAs. HEK293T cells either undepleted or depleted of LC3B were transiently transfected with plasmids expressing RLuc and FLuc reporter mRNAs. The cells were either treated or not treated with Rapa + CQ for 12 h before cell harvest. The amounts of the RLuc mRNAs were normalized to those of FLuc mRNAs. Then, the normalized levels in the untreated cells were arbitrarily set to 100%; *n* = 3; Data are presented as mean values ± SD; *p* values were analyzed using two-tailed and equal-variance Student's *t* test; *\*\*p* < 0.01 (The exact *p* values are provided in Source Data file). **e** Half-life measurement of the RLuc reporter mRNA, RLuc-P3′-WT. As performed in **d** except that the cells were harvested at the indicated time points. The *y* axis represents the level of mRNA remaining (percentage) on the logarithmic scale (log2); *n* = 4; Data are presented as mean values ± SD. **f**, **g** Half-life of endogenous LMD substrates after treatment with Rapa + CQ or LC3B downregulation. Two endogenous LMD substrates, *PRMT1* mRNA (**f**) and *MARS1* mRNA (**g**) were analyzed. *n* = 3; Source data are provided as a Source Data file.

that endogenous LC3B was more strongly associated with RLuc-P3′-WT mRNA than with RLuc-P3′-U/A mRNA (Fig. 3c and Supplementary Fig. 6a). In addition, treatment with Rapa + CQ increased the association between LC3B and RLuc-P3′-WT mRNA by ~1.7-fold. In agreement with these observations, endogenous *PRMT1* mRNA and *MARS1* mRNA (another LMD substrate) were enriched in in vivo CLIP of endogenous LC3B by ~2-fold more, and this enrichment was further increased by ~1.5-fold upon treatment with Rapa + CQ (Supplementary Fig. 6b). Furthermore, endogenous *PRMT1* mRNA and *MARS1* mRNA were efficiently enriched in *the* in vivo CLIP of proLC3B-WT, but not proLC3B-R/Q (Supplementary Fig. 6c, d). These data indicate that LC3B directly binds to AAUAAA in the 3′UTR of *PRMT1* mRNA and that its binding to AAUAAA is increased upon autophagic activation. It should be noted that we observed a differential protein migration pattern upon SDS-PAGE when samples of wild-type and R/Q mutant of proLC3B were used. A previous study reported the same phenomenon[25]. In addition, purified recombinant LC3B-R/Q migrated faster migration than LC3B-WT (Supplementary Fig. 2a). Currently, we do not know the exact reasons for this difference. Probably, a change in protein charge after introducing substitution may affect the migration pattern of proLC3B and efficiency of proLC3B conversion.

Efficient LMD of *PRMT1* mRNA via LC3B binding was corroborated by a decrease in both the abundance and half-life of RLuc-P3′-WT mRNA, but not of RLuc-P3′-U/A mRNA, upon treatment with Rapa + CQ (Fig. 3d, e and Supplementary Fig. 6e, f). The observed decrease was almost completely reversed by LC3B downregulation. The half-life of endogenous *PRMT1* mRNA and *MARS1* mRNA also decreased upon treatment with Rapa + CQ and reversed after LC3B downregulation (Fig. 3f, g). Collectively, our observations indicate that *PRMT1* mRNA, an encoding protein of which functions as a negative regulator of autophagy, is a bona fide substrate for LMD.

**Efficient LC3B-binding to LMD substrates depends on secondary or tertiary structures upstream of AAUAAA motif.** Our CLIP-seq data showed that not all AAUAAA motifs were bound by LC3B, even though most mRNAs had a PAS. We identified two molecular characteristics of the 3′UTR involved in the efficient binding of LC3B to the AAUAAA motif at the transcriptome level (Fig. 4a). First, the mRNAs harboring the 3′UTR AAUAAA motif that binds to LC3B (3′UTR_AAUAAA group) tended to form a long-range base-pairing between the region immediately downstream of the AAUAAA motif and the region immediately downstream of a translation termination codon (Fig. 4a). Although long-range looping was disrupted within the 3′UTR of RLuc-P3′-WT mRNA (see RLuc-P3′-M1 mRNA and RLuc-P3′-M2 mRNA), the mRNA abundance was still efficiently reduced upon treatment with Rapa + CQ (Supplementary Fig. 7a–c). This suggests that the predicted interaction within the 3′UTR may not contribute to efficient LMD. Second, there was a strong indication that the 3′UTR_AAUAAA group had a structured region with a high GC content upstream of the AAUAAA motif, as evidenced by the lower minimum free energy (MFE) of the 3′UTR_AAUAAA group than that of the non-targets (Fig. 4b). Of note, the 3′UTR of *PRMT1* mRNA identified as a bona fide LMD substrate also had a stable hairpin structure with a high GC content (Supplementary Fig. 7d, e).

The role of the stable hairpin structure in LMD was further validated through in vivo CLIP experiments using α-LC3B antibody and qRT-PCR using RLuc-P3′-WT mRNA, RLuc-P3′-M3 mRNA (which harbors seven nucleotide substitutions in the stem region and consequently disrupts the hairpin structure), or RLuc-P3′-M4 mRNA (which harbors an additional seven nucleotide substitutions in the complementary strand in the M3 backbone and consequently restores the stable hairpin structure) (Fig. 4c). The results of the in vivo CLIP experiments revealed that LC3B efficiently interacted with WT and M4 mRNAs, but not with M3 mRNA (Fig. 4d and Supplementary Fig. 7f). In agreement with the finding that autophagic activation increases the association between LC3B and RNA (Fig. 3c and Supplementary Fig. 6b, d), treatment with Rapa + CQ increased the amount of coimmunoprecipitated WT and M4 mRNAs by 2.2- and 1.8-fold, respectively (Fig. 4d). Furthermore, treatment with Rapa + CQ caused efficient degradation of WT and M4 mRNAs, but not of M3 mRNA (Fig. 4e). All these data indicate that a secondary or tertiary structure present upstream of the AAUAAA motif helps to load LC3B onto mRNA, thereby leading to efficient LMD.

**Efficient LMD requires an RNA-binding ability and PE conjugation of LC3B.** To further investigate the molecular aspects of LMD, we carried out complementation experiments using *LC3B* siRNA (which targets the 3′UTR of endogenous *LC3B*) and FLAG-tagged LC3B or its variants (Fig. 5a): FLAG-proLC3B (a precursor form of LC3B), LC3B (a cleaved form of proLC3B by ATG4) harboring either WT or R/Q mutant, respectively, or LC3B-G120A, which fails to conjugate with PE due to substitution of its C-terminal glycine to alanine[33].

Specific downregulation of endogenous LC3B and proper expression of FLAG-LC3B variants were validated using western blotting with α-LC3B antibody or α-FLAG antibody (Fig. 5b, d). The levels of RLuc-P3′-WT mRNA (Fig. 5c, e) and endogenous LMD substrates (*PRMT1* mRNA and *MARS1* mRNA; Supplementary Fig. 8a–d) were reduced by ~2-fold upon treatment with Rapa + CQ and almost completely restored by the expression of either proLC3B-WT or LC3B-WT, but not proLC3B-R/Q, LC3B-R/Q, or LC3B-G120A. Accordingly, the protein levels of endogenous PRMT1 were positively correlated with the levels of *PRMT1* mRNA under these conditions (Fig. 5b, d). Notably, we also obtained comparable results when autophagy was induced by serum deprivation (Supplementary Fig. 9), suggesting that LMD is a general mechanism observed during autophagy. Collectively, these results indicate that both the RNA-binding ability and conjugation of LC3B to PE are necessary for efficient LMD upon autophagic activation.

When and where does LC3B bind to LMD substrates within cells? To address these questions, whole-cell extracts were fractionated into nuclear and cytoplasmic fractions. While U1 snRNP70 (a nuclear protein) and GAPDH (a cytoplasmic protein) were exclusively enriched in the nuclear and cytoplasmic fractions, respectively, both LC3B-I and LC3B-II were largely detected in the cytoplasmic fraction (Fig. 5f). Notably, the amounts of RLuc-P3′-WT mRNA (Fig. 5g) and endogenous LMD substrates (*PRMT1* mRNA and *MARS1* mRNA; Supplementary Fig. 8e, f) were largely reduced in the cytoplasm in an LC3B-dependent manner. In addition, in agreement with our observations that efficient LMD requires the conversion of LC3B-I to LC3B-II (Fig. 5d, e), the LMD of RLuc-P3′-WT mRNA was almost completely inhibited by treatment with 3-methyladenine (3-MA), which inhibits phosphatidylinositol 3-kinases and consequently blocks autophagosome formation (Fig. 5h). In contrast, treatment with bafilomycin A1 (Baf A1), CQ (both known to block the fusion of autophagosomes to lysosomes, consequently inhibiting a late step of autophagy), or a potent translation inhibitor (puromycin or cycloheximide) had no significant effect on LMD efficiency (Fig. 5h). These data suggest

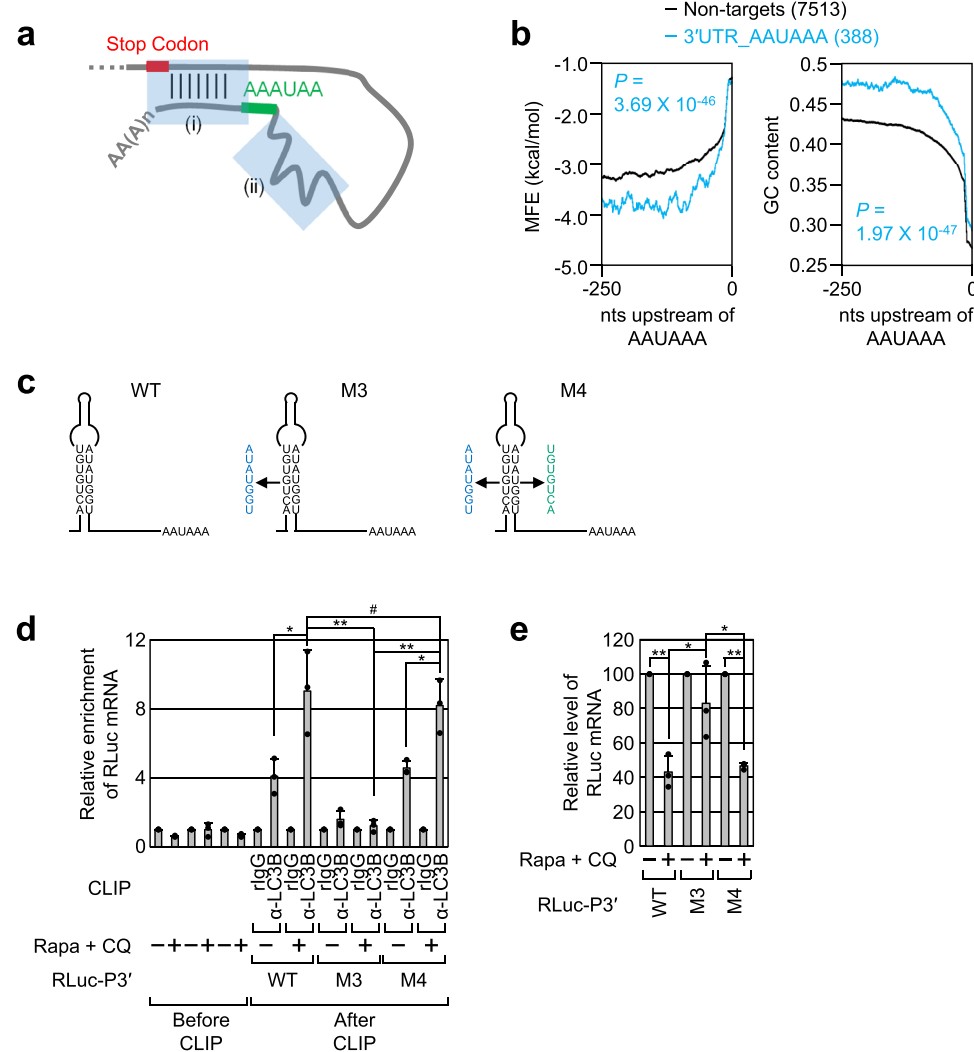

**Fig. 4 Presence of secondary or tertiary structures upstream of the AAUAAA motif promotes an association between LC3B and LMD substrates. a** Common features found in LMD substrates. Two common features, (i) a long-range looping and (ii) the secondary or tertiary structures upstream of the AAUAAA motif, are indicated by semi-transparent rectangles. **b** MFE values (left) and GC content (right) across the nucleotide positions upstream of the AAUAAA motif. The 3′UTR sequences of mRNAs belonging to the 3′UTR_AAUAAA or non-targets group were analyzed. *p* values were calculated using the two-tailed Kolmogorov–Smirnov test. **c** Predicted secondary structure of the 3′UTR of PRMT1-WT, M3, and M4. **d** In vivo CLIPs of endogenous LC3B. As performed in Fig. 3c, except that HEK293T cells were expressed with one of RLuc-P3′ reporter mRNAs (WT, M3, or M4 mRNA). *n* = 3; Data are presented as mean values ± SD; *p* values were analyzed using two-tailed and equal-variance Student's *t* test; # not significant; *$p < 0.05$; **$p < 0.01$ (The exact *p* values are provided in Source Data file). **e** Relative amounts of the RLuc-P3′ reporter mRNAs upon treatment with Rapa + CQ. *n* = 3; Data are presented as mean values ± SD; *p* values were analyzed using two-tailed and equal-variance Student's *t* test; *$p < 0.05$; **$p < 0.01$ (The exact *p* values are provided in Source Data file). Source data are provided as a Source Data file.

that efficient LMD occurs mostly in the cytoplasm in a translation-independent manner before the formation of autolysosomes.

**LC3B elicits a rapid decay of LMD substrates via a CCR4-NOT complex.** How does LC3B elicit the rapid degradation of LMD substrates? Our efforts to answer this question revealed that CNOT1, which is the largest component of the CCR4-NOT deadenylase complex and functions as a scaffold protein, coimmunopurified with endogenous LC3B (Fig. 6a). Treatment with Rapa + CQ increased the amount of coimmunoprecipitated CNOT1 and CNOT7 (which has a deadenylase ability as a component of the CCR4-NOT complex) by ~2-fold in the IPs of LC3B. Notably, downregulation of ATG5 and ATG7 reduced the levels of coimmunoprecipitated CNOT1 and CNOT7 (Fig. 6b, c), suggesting that the conversion of LC3B-I to LC3B-II increases the

association between LC3B and the CCR4-NOT deadenylase complex. Furthermore, using a proximity ligation assay (PLA), which allows the detection of protein-protein interactions within the cells with high specificity and sensitivity[34], we observed distinct PLA spots (PLA signals) between endogenous LC3B and CNOT1 and between endogenous LC3B and CNOT7 (Fig. 6d and Supplementary Fig. 10). The number of PLA spots increased upon treatment with Rapa + CQ. The functional relevance of CCR4-NOT in LMD was demonstrated by observations that (1) downregulation of LC3B, CNOT1, or CNOT7 increased the relative levels of RLuc-P3′-WT reporter mRNA and endogenous LMD substrates (*PRMT1* mRNA and *MARS1* mRNA; Fig. 6f), and (2) Rapa treatment shortened the length of poly(A) of endogenous LMD substrates (*COTL1* mRNA and *CCT7* mRNA) in a way that was reversed by LC3B downregulation. (Supplementary Fig. 11).

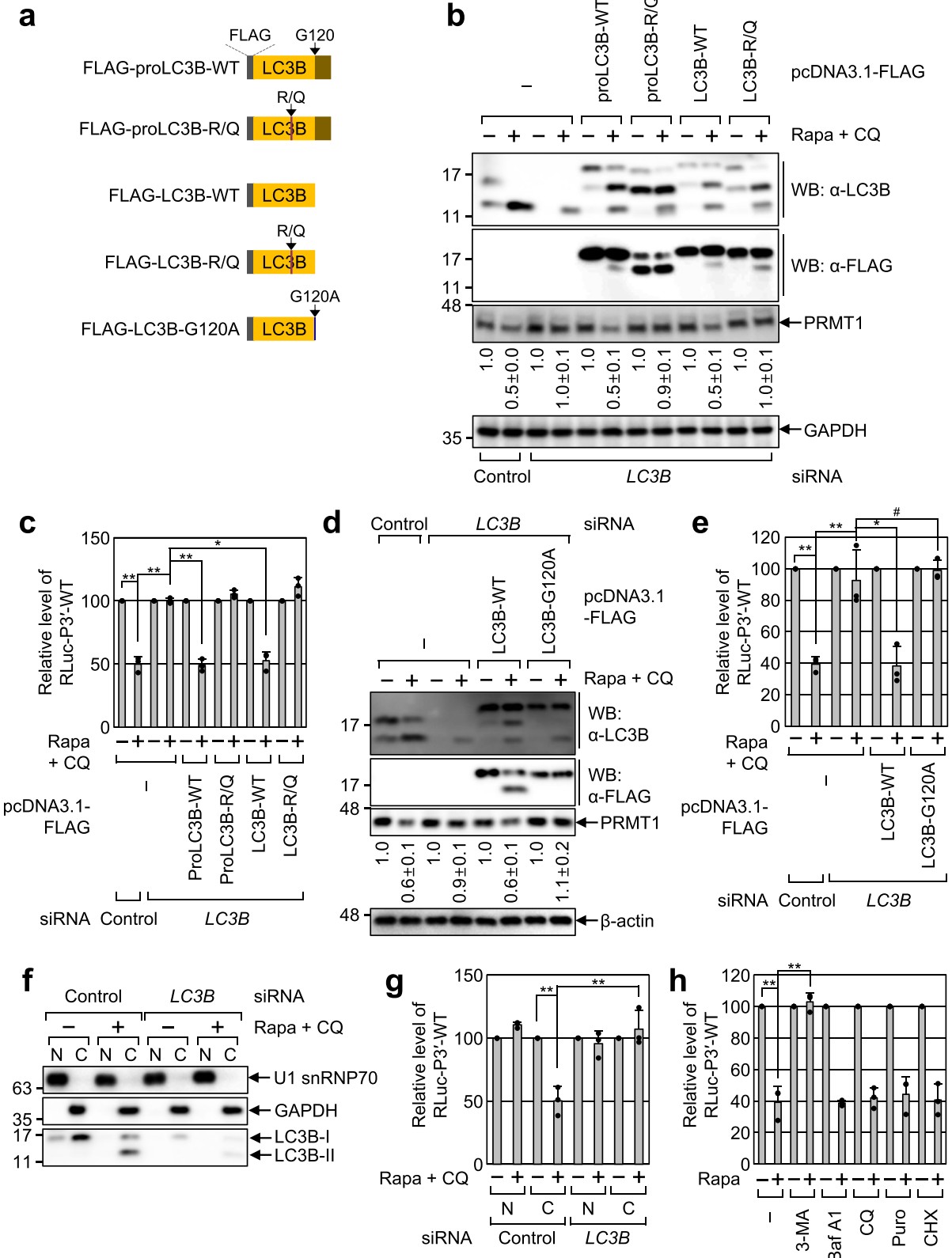

As mentioned above in Fig. 5, we proposed that LMD takes place before the formation of autolysosomes. Therefore, it is expected that the increased association between LC3B and the CCR4-NOT complex after autophagic activation may be observed at or around the autophagic puncta. Accordingly, FLAG-LC3B and endogenous p62 (another marker for autophagy puncta) were significantly enriched in autophagic puncta upon treatment

with Rapa + CQ; however, the PLA spots of LC3B and either CNOT1 or CNOT7 partially overlapped with autophagic puncta (Fig. 7a and Supplementary Fig. 12). In addition, the observed PLA spots did not colocalize with DCP1A (a marker for processing bodies) or G3BP1 (a marker for stress granules). Notably, we also observed that, upon autophagic activation, endogenous CNOT1 and CNOT7 coimmunopurified with ATG5,

**Fig. 5 Efficient LMD involves an RNA-binding ability and PE conjugation of LC3B. a** Schematic diagram of FLAG-LC3B variants used in this study. **b–e** Complementation experiments using LC3B variants. HEK293T cells were transiently transfected with either the control siRNA or *LC3B* siRNA that anneals to 3′UTR of endogenous *LC3B* mRNA. Two days later, the cells were retransfected with an RLuc-P3′-WT reporter plasmid, a FLuc reference plasmid, and a plasmid expressing either FLAG-LC3B-WT or its variant. The cells were either treated or not treated with Rapa + CQ for 12 h before cell harvest. **b, d** Western blotting showing specific downregulation of endogenous LC3B and proper expression of FLAG-LC3B or its variant at a level comparable to that of endogenous LC3B. **c, e** Effect of LC3B-WT or its variant on LMD. The amounts of RLuc mRNAs were normalized to those of FLuc mRNAs. Then, the normalized levels in the untreated cells were arbitrarily set to 100%. $n = 3$; # not significant; *$p < 0.05$; **$p < 0.01$. **f, g** Measurement of LMD efficiency in the nucleus and cytoplasm. **f** Western blotting showing specific separation between the nuclear (N) and cytoplasmic (C) fractions. **g** Efficiency of LMD of RLuc-P3′-WT reporter mRNAs. $n = 3$; **$p < 0.01$. **h** Effect of treatment with the indicated chemical inhibitor on LMD of RLuc-P3′-WT reporter mRNAs. $n = 3$; **$p < 0.01$. In **c, e, g, h**, data are presented as mean values ± SD from three biological replicates. $p$ values were analyzed using two-tailed and equal-variance Student's $t$ test; The exact $p$ values are provided in Source Data file. Source data are provided as a Source Data file.

---

ATG12, and ATG16L1, all of which are enriched in the autophagophore (Fig. 7b). Taken together with our observations that LMD occurs before the formation of autolysosomes, these data suggest that the increased association between LC3B and the CCR4-NOT complex after autophagic activation is initiated outside of autophagic puncta.

**Efficient LMD of *PRMT1* mRNA facilitates autophagy.** Based on a previous observation that PRMT1 acts as a negative regulator of autophagy[32], we next investigated the functional importance of *PRMT1* mRNA degradation via LMD during autophagy. We generated several reporter mRNAs (Fig. 8a) harboring a FLAG-tagged full-length *PRMT1* ORF resistant to siRNA and the full-length 3′UTR of *PRMT1* mRNA (WT, M3, or M4 depicted in Fig. 4). Following treatment with Rapa + CQ, downregulation of PRMT1 increased the number of LC3B puncta per cell (Fig. 8b–d), which is consistent with the role of PRMT1 as a negative regulator of autophagy. FLAG-PRMT1[R]-WT or its variant was expressed under the same conditions. In agreement with our findings that efficient LMD requires a secondary or tertiary structure to facilitate efficient loading of LC3B onto AAUAAA (Fig. 4), FLAG-PRMT1[R]-WT and -M4 mRNA, but not FLAG-PRMT1[R]-M3 mRNA, were efficiently degraded by LMD (Fig. 8b). Accordingly, the levels of the FLAG-PRMT1 protein were positively correlated with those of the FLAG-PRMT1[R] reporter mRNA (Supplementary Fig. 13a). Notably, the expression of FLAG-PRMT1[R]-WT and -M4 mRNA did not significantly affect the number of LC3B puncta; however, the expression of FLAG-PRMT1[R]-M3 mRNA significantly reduced the number of LC3B puncta per cell (Fig. 8c, d). We also repeated similar complementation experiments under Rapa-treated conditions or serum-depleted conditions, then measured the protein levels of FLAG-PRMT1 and p62 (the latter of which represents autophagic turnover). FLAG-PRMT1 and p62 were efficiently reduced when PRMT1-depleted cells were expressed with PRMT1 FLAG-PRMT1[R]-WT or -M4 mRNA, but not with FLAG-PRMT1[R]-M3 mRNA under both conditions (Supplementary Fig. 13b, c). Furthermore, the protein levels of endogenous PRMT1 and p62 were reduced by 2.5-fold and ~10-fold under Rapa treatment, respectively, and they were significantly restored by downregulation of CNOT1 or CNOT7 (Supplementary Fig. 13d). Collectively, these data indicate that rapid degradation of *PRMT1* mRNA through the direct binding of LC3B to *PRMT1* 3′UTR facilitates autophagy.

## Discussion

Many RBPs are known to be involved in a variety of molecular and cellular processes[11–14,26,27]. Although LC3, a central protein in autophagy, has been implicated in RNA-binding ability, the potential effect of this ability on autophagy remains unknown. In this study, we clearly demonstrate that LC3B directly binds to a

subset of mRNAs harboring the AAUAAA consensus motif, resulting in rapid degradation of the mRNAs upon autophagic activation. Furthermore, we show that efficient degradation of *PRMT1* mRNA via LMD facilitates autophagy. Therefore, our observations suggest that, upon autophagic activation, LMD contributes to the creation of a suitable intracellular environment for efficient autophagy activity (degradation of cargo via formation of autolysosomes) by specifically degrading its target substrates before the formation of autolysosomes (Fig. 8e).

With respect to the molecular mechanism, LMD is a translation-independent and inducible mRNA decay pathway (Fig. 5h), which is mechanistically reminiscent of glucocorticoid receptor-mediated mRNA decay induced by the presence of a specific ligand[35–37]. Under normal conditions, LC3B preferentially associates with target mRNAs harboring the AAUAAA motif (Fig. 1). Although general RBPs bind to their target RNAs with dissociation constant ranging from nanomolar to micromolar values[38], our fluorescence polarization assay showed a low RNA-binding affinity of purified recombinant LC3B to the AAUAAA motif (Fig. 1f). However, the association and specific loading of LC3B onto target mRNAs within cells is expected to be guided by RNA structures and other RBPs upstream of the AAUAAA motif (Fig. 4). Upon autophagic activation, LC3B is conjugated to PE. Concomitantly, autophagic activation increases LC3B binding to target mRNAs in a way that is affected by RNA structures and other RBPs located upstream of the LC3B-binding site (Fig. 4). Therefore, it is plausible that LC3B conversion not only causes intracellular redistribution toward the autophagic puncta of LC3B, but also reinforces target recognition for LMD.

We also observed that autophagic activation leads to an increased association between LC3B and the CCR4-NOT complex (Fig. 6a–d). This event occurs either independently of or before the formation of LC3B puncta, as evidenced by our PLA data showing that the interactions between endogenous LC3B and CNOT1 (or CNOT7) occur throughout the cells (Supplementary Fig. 10). The recruited CCR4-NOT complex might trigger the rapid deadenylation of LMD substrates in the cytoplasm. Alternatively, considering that (1) the PLA signals between LC3B and CNOT1 and between LC3B and CNOT7 partially overlap with LC3B or p62 puncta (Fig. 7a and Supplementary Fig. 12), and (2) endogenous CNOT1 and CNOT7 associate with ATG5, ATG12, and ATG16L1 upon autophagic activation (Fig. 7b), the complex comprising mRNA, LC3B-II, and CCR4-NOT may move to the autophagophore, and CCR4-NOT may elicit rapid deadenylation of LMD substrates on the surface of the autophagophore. In either case, the deadenylated LMD substrates would be completely degraded by exosome-mediated 3′-to-5′ exoribonucleolytic cleavage and/or decapping followed by XRN1-mediated 5′-to-3′ exoribonucleolytic cleavage[37].

In this study, we propose that the rapid degradation of *PRMT1* mRNA via LMD facilitates autophagy. However, in addition to *PRMT1* mRNA, LMD may target a variety of cellular mRNAs

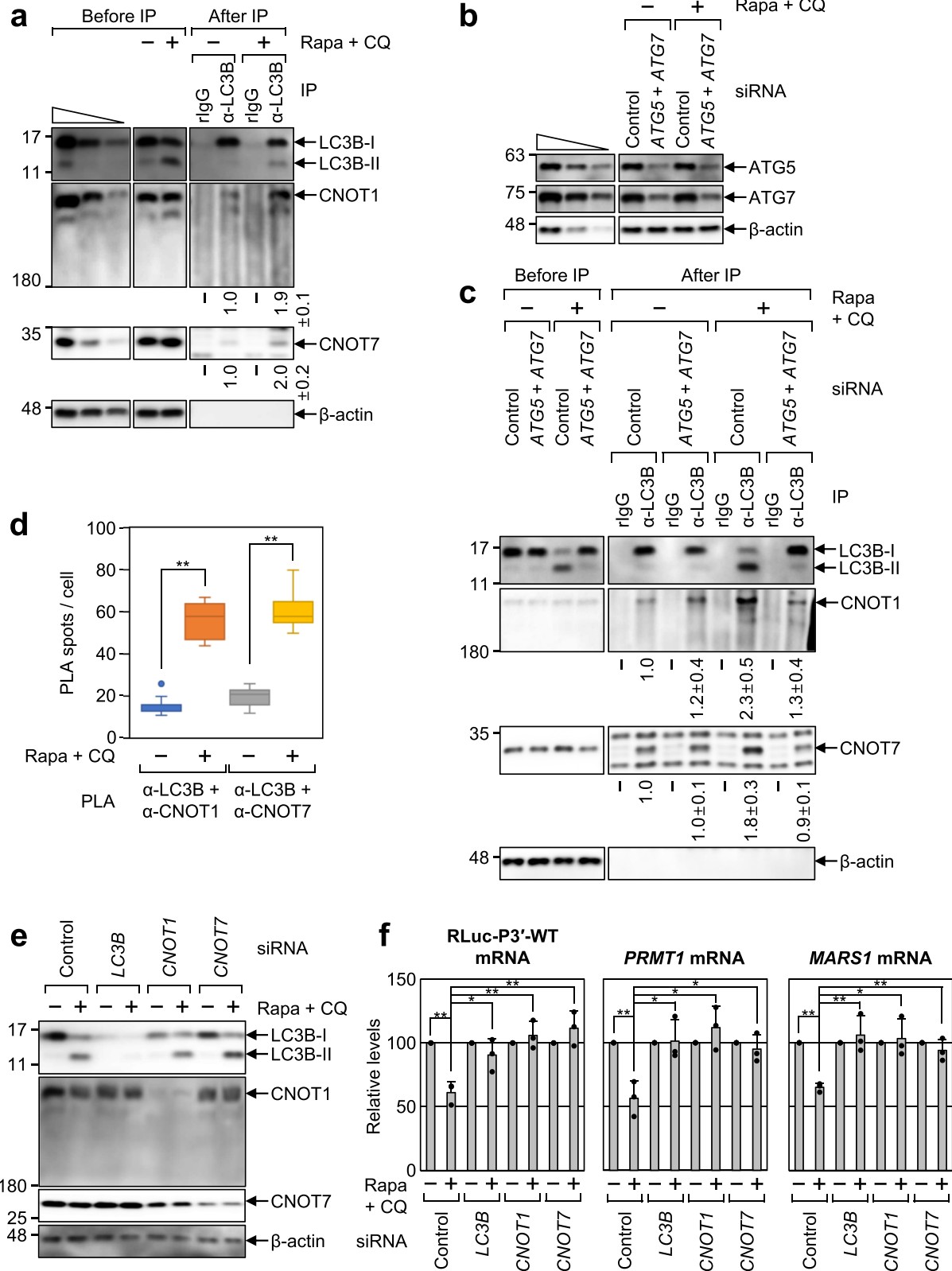

involved in autophagy. For instance, DNA damage-inducible transcript 4 (DDIT4), which is known to repress mTORC1-mediated signaling[39], was identified as an LMD substrate in our transcriptomic analysis (Supplementary Fig. 5). A recent study showed that DDIT4 induction leads to excessive release of reactive oxygen species, resulting in impaired autophagy in dry eye disease[40]. Therefore, proper coordination of various LMD

substrates before autolysosome formation might be a prerequisite for efficient autophagy.

Molecular and cellular functions via the RNA-binding ability of LC3B may not be limited to conventional autophagy. Recent accumulating evidence suggests that the core components involved in autophagy also mediate various autophagy-independent pathways or noncanonical autophagy[41,42]. In

**Fig. 6 CCR4-NOT complex associates with LC3B, promoting efficient LMD. a** IPs of endogenous LC3B. The extracts of HEK293T cells either treated or not treated with Rapa + CQ were subjected to IPs using α-LC3B antibody or nonspecific rIgG. The intensities of each western blot image were quantitated. The intensities of coimmunoprecipitated proteins were normalized to those of immunoprecipitated LC3B. The relative levels obtained in untreated cells were arbitrarily set to 1.0. $n = 3$. **b, c** IPs of endogenous LC3B upon downregulation of both ATG5 and ATG7. As performed in **a**, except that the cells were either undepleted or depleted of ATG5 and ATG7. **b** Western blotting proving specific downregulation of endogenous ATG5 and ATG7. **c** Western blotting of cellular proteins before or after IPs of LC3B. **d** The proximity ligation assay (PLA) between endogenous LC3B and either CNOT1 or CNOT7. PLA experiments using the indicated antibodies were performed on HeLa cells treated with either DMSO or Rapa + CQ. The PLA images are shown in Supplementary Fig. 10. The number of PLA spots per cell was quantified and is presented in this panel. Box-whiskers show maximum, third quartile to first quartile, median and minimum; $n = 767$ cells examined over three independent experiments. **e, f** Effect of CNOT1 or CNOT7 downregulation on LMD. **e** Western blotting showing specific downregulation of endogenous CNOT1 and CNOT7. **f** LMD efficiency of RLuc-P3′-WT reporter mRNA, endogenous *PRMT1* mRNA, and *MARS1* mRNA. $n = 3$; Data are presented as mean values ± SD; $p$ values were analyzed using two-tailed and equal-variance Student's $t$ test; *$p < 0.05$; **$p < 0.01$ (The exact $p$ values are provided in Source Data file). Source data are provided as a Source Data file.

---

particular, LC3B is known to function in phagocytosis, exocytosis, cytokine secretion, pathogen inclusion, and viral replication in an autophagy-independent or noncanonical autophagic manner[43–46]. Remarkably, our gene ontology analysis showed that putative LMD substrates were significantly enriched in distinct GO terms, including the viral process (Supplementary Fig. 14). Therefore, further studies on the possible interplay between the RNA-binding ability of LC3B and various autophagy-independent pathways are required to broaden our knowledge of the biological roles of LMD.

## Methods

**Plasmid construction.** The following plasmids were purchased or described previously: RLuc (Promega, originally named as pRL-CMV), pcDNA3-FLAG-DCP1A[47], pCI-F, pcDNA3-FLAG[36], and pCNS-D2-PRMT1 (BKU006191, purchased from 21C Frontier Human Gene Bank, KRIBB, Korea).

pcDNA3.1-FLAG was constructed by inserting a FLAG tag into pcDNA3.1 (Invitrogen; V79520).

pcDNA3.1-FLAG-proLC3B-WT was constructed by inserting a complementary DNA (cDNA) fragment encoding full-length proLC3B into pcDNA3.1-FLAG.

pcDNA3.1-FLAG-proLC3B-R/Q, which harbors three amino acid substitutions from arginine to glutamine in arginine-rich motifs (amino acids 68, 69, and 70 residues of LC3B), was constructed using a two-step polymerase chain reaction (PCR). The 5′ and 3′ fragments were amplified by PCR using pcDNA3.1-FLAG-proLC3B as a template and specific oligonucleotides: (1) 5′-GACTCACTATAG GGAGACCCAAGCTGGC-3′ (sense) and 5′-GAGCTGTAATTGCTGTTGAAT TATCTTGATGAGCTCAC-3′ (antisense) for amplification of the 5′ fragment and (2) 5′-CAAGATAATTCAACAGCAATTACAGCTCAATGCTAATCAG-3′ (sense) and 5′-CAAACAACAGATGGCTGGCAACTAGAAG-3′ (antisense) for amplification of the 3′ fragment, where the underlined nucleotides specify the substitution sites. The two PCR-amplified fragments were mixed and reamplified by PCR using the sense oligonucleotide used for amplification of the 5′ fragment and the antisense oligonucleotide used for amplification of the 3′ fragment. The resulting PCR-amplified fragment was digested with XhoI and HindIII and ligated to an XhoI/HindIII fragment of pcDNA3.1-FLAG-proLC3B. The other plasmids expressing an LC3B variant, pcDNA3.1-FLAG-LC3B, pcDNA3.1-FLAG-LC3B-R/Q, and pcDNA3.1-FLAG-LC3B-G120A were also generated by two-step PCR.

RLuc-P3′-WT, -U/A, -M1, -M2, -M3, or -M4, which harbor the entire 3′UTR of *PRMT1* mRNA or its variant, was constructed by inserting an in vitro-synthesized DNA fragment harboring the entire 3′UTR or its variant sequences of the *PRMT1* mRNA immediately downstream of the RLuc gene.

To generate pcDNA3-FLAG-PRMT1[R]-WT, -M3, and -M4, which express the *PRMT1* siRNA-resistant form of *PRMT1* mRNA but encode the wild-type PRMT1 protein, pcDNA3-FLAG-PRMT1-WT was first constructed by inserting the full-length *PRMT1* cDNA from pCNS-D2-PRMT1 into pcDNA3-FLAG. Then, siRNA-resistant sequences were inserted into the *PRMT1* ORF using two-step PCR. First, the 5′ half and 3′ half of the fragment encoding PRMT1[R] were amplified by PCR using pcDNA3-FLAG-PRMT1-WT as a template with specific oligonucleotides: (1) 5′-CGCGGATCCGCCGGATGGCGGCAGCCGAGGCCGCG-3′ (sense) and 5′-GAGGTGAAGGTCAGGTCTTCCACC*TTCACCGTGTATATATCGACT*TCCTT-TATGAGGCAG-3′ (antisense) for the amplification of the 5′-half of the fragment encoding PRMT1[R] and (2) 5′-GTCACCAACGCCTGCCTCATAAAGGA*AGTC GATATATACACGGTGAA*GGTGGAAGACCTGAC-3′ (sense) and 5′-CTTGG TACCGAGCTGACCCCAGCTGAGGATTTATTGGAG-3′ (antisense) for the 3′ half of the fragment encoding PRMT1[R], respectively, where the mutated sequences are italicized. Next, the two PCR-amplified fragments were mixed and reamplified by PCR with the sense oligonucleotide used for the amplification of the 5′-half of the fragment and the antisense oligonucleotide employed for amplification of the 3′ half of the fragment. The resulting PCR-amplified fragment was digested with BamHI/Acc65I and then ligated to the BamHI/Acc65I fragment of pcDNA3-FLAG. After that, the 3′UTR sequences were replaced with the DNA fragment

harboring either the full-length or its variant *PRMT1* 3′UTR obtained from RLuc-P3′-WT, -M3, or M4.

**Cell culture.** HEK293T cells (fetal; ATCC), HeLa cells (female; ATCC), and HeLa cells stably expressing GFP-LC3B (a gift from Dr. Chungho Kim, Korea University, Seoul, Korea) were cultured in Dulbecco's modified Eagle's medium (Sigma-Aldrich) containing 10% fetal bovine serum (Sigma-Aldrich) and 1% penicillin/streptomycin (Sigma-Aldrich). To prevent mycoplasma contamination, all the cultured cells were regularly treated with Plasmocin™ (Invivogen) and examined using the MycoAlert PLUS Mycoplasma Detection Kit (Lonza).

Where indicated, the cells were treated with rapamycin (Rapa; 500 nM for HEK293T cells and 100 μM for HeLa cells; Enzo Life Science) predissolved in DMSO (BioShop) with or without 30 μM chloroquine (CQ; Sigma-Aldrich) for 12 h. In addition, the cells were treated with 10 mM 3-MA (Sigma-Aldrich), 100 nM Baf A1 (Calbiochem), 1 μg/ml puromycin (Puro; Sigma-Aldrich), or 50 μg/ml cycloheximide (CHX; Sigma-Aldrich) for 12 h.

**DNA or siRNA transfection.** For IP and in vivo CLIP experiments, HEK293T cells were transiently transfected with the indicated plasmids using the calcium phosphate method. For reporter assays and confocal microscopy, cells were transiently transfected with the indicated plasmids using Lipofectamine 2000. Two days after DNA transfection, the cells were harvested or fixed.

To downregulate endogenous proteins, the cells were transfected with 100 nM in vitro-synthesized siRNA (Gene Pharma) using Lipofectamine 3000 (Invitrogen). Three days after siRNA transfection, cells were harvested or fixed. The siRNA sequences used in this study are listed in Supplementary Data 3a.

**Antibodies.** Primary antibodies against the following proteins were purchased [listed in the format "protein name (catalog number, supplier)"]: LC3B (NB100-2220, Novus; #3868, Cell Signaling Technology; or AM20212PU-N, ORIGENE), CNOT1 (14276-1-AP, Proteintech), CNOT7 (ab195587, Abcam), PRMT1 (07-404, Sigma-Aldrich), FLAG (A8592, Sigma-Aldrich; or F3165, Sigma-Aldrich), p62 (Cell Signaling Technology; 18420-1-AP), G3BP1 (13057-2-AP, Proteintech), GFP (sc-9996, Santa Cruz Biotechnology), U1 snRNP70 (sc-390899, Santa Cruz Biotechnology), β-actin (A5441, Sigma-Aldrich), and GAPDH (LF-PA0212, Ab Frontier).

The following secondary antibodies against the primary antibodies were used in this study: peroxidase-conjugated goat anti-mouse IgG antibody (AP124P, Sigma-Aldrich) and peroxidase-conjugated goat anti-rabbit IgG antibody (AP132P, Sigma-Aldrich) for western blotting, and Alexa Fluor ® 488-conjugated goat anti-mouse IgG antibody (A11017, Invitrogen) and rhodamine-conjugated goat anti-rabbit IgG antibody (31670, Invitrogen) for immunostaining.

**Cross-linking immunoprecipitation followed by sequencing.** HEK293T cells were washed with ice-cold PBS and irradiated with ultraviolet light (UV; 254 nm, 400 mJ/cm$^2$) using a UV cross-linker CL-1000 (UVP) before harvesting. The cross-linked cell pellets were resuspended in 600 μl of lysis buffer [20.3 mM Na$_2$HPO$_4$, 3.5 mM KH$_2$PO$_4$, 6.8 mM KCl, 342.5 mM NaCl (pH 7.2), 0.1% SDS, 0.5% sodium deoxycholate, 0.5% NP-40, 10 mM sodium fluoride, and 0.25 mM sodium ortho-vanadate] and incubated for 10 min on ice. Then, the cell lysates were mixed with RNase-free DNase I (30 U; Thermo Scientific), incubated for 5 min at 37 °C, mixed with 0.1 or 1 ng of RNase A (Affymetrix), and incubated for additional 10 min at 37 °C. The total cell extracts were subjected to centrifugation, and the supernatants were mixed with protein A Dynabeads (Invitrogen) preincubated with the primary antibody for 2 h at 4 °C. The beads were then washed twice with lysis buffer and twice with PNK buffer [50 mM Tris-HCl (pH 7.4), 10 mM MgCl$_2$, and 0.5% NP-40].

To remove the phosphate groups, the beads were incubated with 40 U/ml of alkaline phosphatase (Roche) at 37 °C for 10 min. The beads were then washed twice with PNK + EGTA buffer [50 mM Tris-HCl (pH 7.4), 0.5% NP-40, and 20 mM EGTA] and twice with PNK buffer. To phosphorylate, the beads were

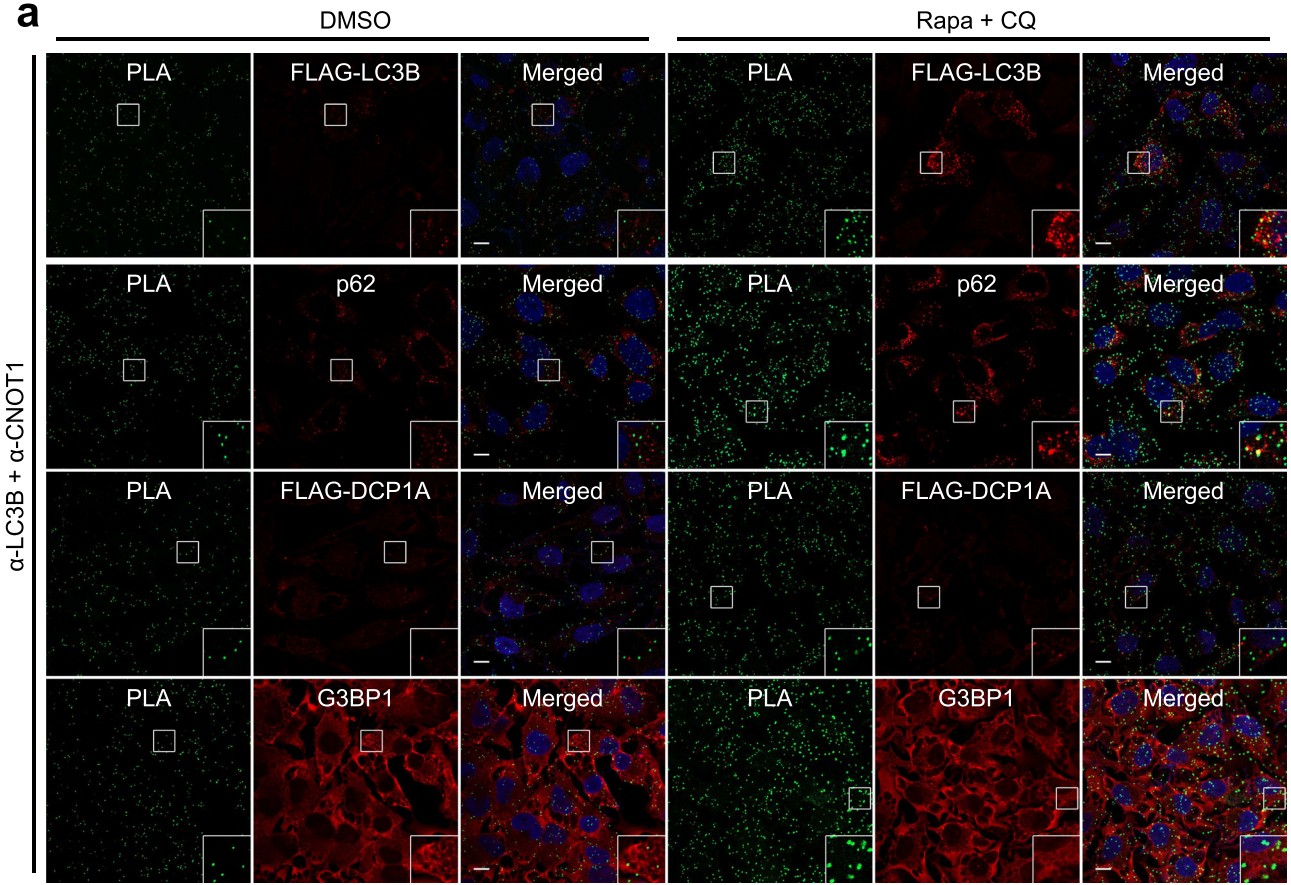

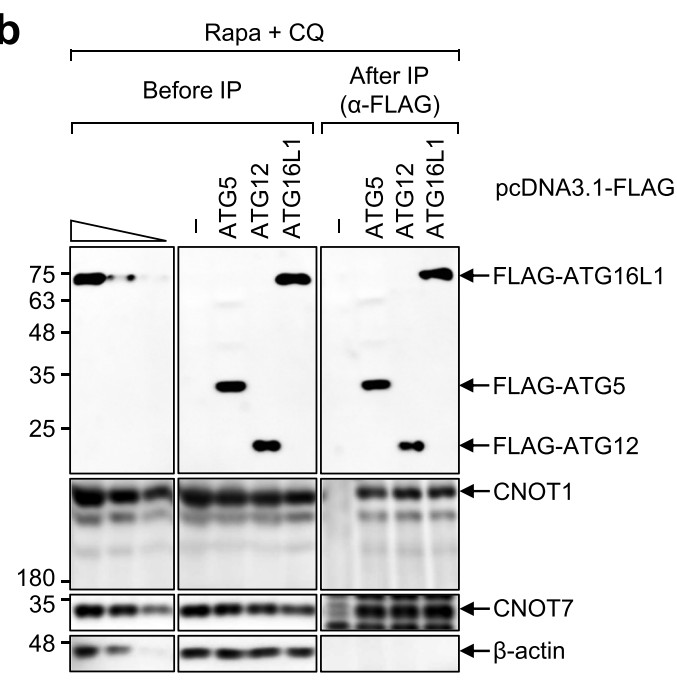

incubated with 0.5 U/µl T4 PNK (New England Biolabs) and 0.125 mM ATP for 15 min at 37 °C and washed twice with PNK + EGTA buffer. Finally, RNAs and proteins bound to the beads were eluted using a lithium dodecyl sulfate sample buffer (Invitrogen).

To isolate RNA covalently cross-linked to proteins, the samples were subjected to SDS-PAGE in a Novex NuPAGE Bis-Tris gel (Invitrogen), and the separated proteins were transferred to a nitrocellulose membrane. The membranes

corresponding to ~20 kDa above the molecular weight of the immunoprecipitated proteins were excised, and the membrane was soaked in 4 mg/ml proteinase K (Roche) for 20 min at 37 °C with PK buffer [100 mM Tris-HCl (pH 7.4), 50 mM NaCl, and 10 mM EDTA]. After that, PK/7 M urea buffer [100 mM Tris-HCl (pH 7.4), 50 mM NaCl, 10 mM EDTA, and 7 M urea] was added and incubated for 20 min at 37 °C. The soluble fraction was subjected to organic extraction and ethanol precipitation with GlycoBlue™ (Ambion) to precipitate RNA fragments.

**Fig. 7 Association between LC3B and the CCR4-NOT complex is initiated outside of autophagic puncta. a** Intracellular distributions of PLA spots involving endogenous LC3B and CNOT1. Intracellular distributions of PLA spots showing the specific interaction between endogenous LC3B and CNOT1 were determined using PLA experiments. In addition, the intracellular distributions of FLAG-LC3B and endogenous p62 (both of which were used for markers for autophagosome), FLAG-DCP1A (a marker for processing bodies), or endogenous G3BP1 (a marker for stress granules) were determined by immunostaining with α-FLAG antibody, α-p62 antibody, α-G3BP1 antibody, and α-G3BP1 antibody, respectively. Nuclei were stained with 4′,6-diamidino-2-phenylindole (DAPI; blue). An enlarged view of the white boxed area is provided in the lower right corner of each image. $n = 3$; Scale bar = 10 μm. **b** IPs of ATG5, ATG12, and ATG16L1. HEK293T cells were transiently transfected with a plasmid expressing FLAG, FLAG-ATG5, FLAG-ATG12, or FLAG16L1. Two days later, the cells were treated with Rapa + CQ for 12 h before cell harvest. Total cell extracts were subjected to IPs with the α-FLAG antibody; $n = 2$; Source data are provided as a Source Data file.

Small RNA libraries were constructed using the SMARTer smRNA-Seq Kit for Illumina (Clontech). Sequencing was performed in 50 bp single-end mode using an Illumina HiSeq 2500 system (Illumina, CA, USA) and conducted by Macrogen Inc.

**Preprocessing and mapping of reads**. The raw data obtained from the high-throughput sequencing were preprocessed in the following order: trimming of adapter and poly(A) sequences with Cutadapt[48]. RiboPicker software[49] was used with a customized rRNA dataset to filter out ribosomal sequences from the CLIP-seq data. At the step of adapter-trimming, read lengths over 15 bp with a Phred quality score of ≥30 were chosen for further analysis. The filtered reads were mapped to the reference human genome (hg19) using STAR software[50]. For subsequent analyses, only uniquely mapped reads were selected. Using the NumPy library in Python, Pearson's correlation coefficient ($r$) between two independent biological replicates in each experiment was determined (Supplementary Data 1). The numbers of reads obtained from the stepwise processing are summarized in Supplementary Data 1 (panel a, LC3B CLIP-seq; panel b, mRNA-seq for input RNA in CLIP-seq experiments; panel c, mRNA-seq for the comparison of mRNA abundance; and panel d, mRNA-seq for the measurement of half-life).

**CLIP peak calling**. Peak calling of LC3B CLIP was performed using the piranha software[51] with specific option settings (-z 20 -a 0.9). The mRNA-seq data of the input was used as a covariate. Only overlapping peak centers from two biological replicates were used for further analyses. The exonic regions of the mRNAs were obtained from publicly available databases: RefSeq Genes [hg19; UCSC table browser (http://genome.ucsc.edu/cgi-bin/hgTables)]. Three hierarchical categories were set up according to the following annotation: CDSs, 5′UTRs, and 3′UTRs. Next, the CLIP peaks were assigned to each exonic region using the intersectBed module from Bedtools[52]. The overlapping CLIP peaks from two biological replicates and mRNAs with FPKM ≥ 1 in DMSO-treated cells were used for further analysis.

To determine the distance between the LC3B peaks located at the consensus AAUAAA motif and PAS, random sequences were selected using random permutation modules of the Python NumPy library. The Kolmogorov–Smirnov test in the scipy.stats.ks 2samp module of the SciPy Python library was used for statistical analysis to assess the distributional differences in the distances.

**Metagene analysis**. In the metagene analysis, the longest one among the mRNA isoforms (RefSeq genes) was considered representative. The LC3B CLIP peaks were displayed in their respective regions: 5′UTR, CDS, and 3′UTR. Each region was smoothed using the nearest-neighbor method ($k = 3$) after being binned into 50 segments.

**Consensus motif analysis**. The LC3B CLIP peaks located throughout the mRNA or at the 3′UTR were analyzed using the HOMER package (v.4.8.3)[53] and MEME (v.5.0.1)[54]. The motif length was restricted to six nucleotides in HOMER or six nucleotides in MEME. The predicted motifs with the lowest $p$ or $E$ values were then chosen.

**Analysis of the 3′UTR length change**. To estimate changes in the 3′UTR length after Rapa treatment or LC3B downregulation, an RNA-seq quantification tool of alternative polyadenylation (QAPA version 1.0.0[55]) was used. In brief, a refined set of 3′UTR isoforms was constructed with QAPA from GENCODE (Release 19), and then changes in the length of 3′UTR were determined using the ΔPPAU value.

**mRNA sequencing**. Total cell RNA was purified using TRIzol Reagent. Library construction and mRNA sequencing assays were performed by Macrogen, Inc. Briefly, the RNA integrity number was measured using a bioanalyzer (Agilent). The purified RNA samples were subjected to mRNA-seq library construction using the TruSeq Stranded mRNA Sample Preparation Kit (Illumina). The quality of the constructed cDNA libraries was validated according to the size distribution on an Agilent Bioanalyzer (DNA 1000 kit; Agilent) and quantitated by qPCR (Kapa Library Quant Kit; Kapa Biosystems, Wilmington, MA). The library was adjusted to 2 nmol/l for NGS on the Illumina HiSeq 2500 platform (100-base paired-end reads). To validate the strong correlation between the two replicates, Pearson's

correlation coefficients ($r$) between two biological replicates of mRNA-seq were calculated.

**Profiling of mRNA abundance**. To measure mRNA abundance, the mapped reads from mRNA sequencing were quantified in FPKM using Cufflinks version 2.2.1[56]. The cumulative distribution of mRNA abundance in cells treated or not treated with Rapa was drawn using in-house Python codes. Only genes with FPKM ≥ 1 in mRNA-seq of cells treated with siControl and DMSO were considered. To compute the log2 fold change from Cuffdiff, genes with FPKM = 0 were ignored. The CDF plots for mRNA half-life were plotted using an in-house Python code. The two-tailed Mann–Whitney $U$ test was used to determine $p$ values between targets and non-targets.

**Profiling of mRNA half-life**. For mRNA half-life, HEK293T cells were treated with 100 μg/ml 5,6-dichloro-1-b-Dribofuranosylbenzimidazole (DRB; a potent transcription inhibitor; Sigma-Aldrich) at 0, 6, and 12 h before cell harvesting. Total RNA was purified and qRT-PCR was performed to quantitate mRNA levels at each time point.

For a transcriptome-wide analysis of mRNA half-life, the purified RNAs at each time point were subjected to mRNA sequencing. A customized reference genome, which was defined as the union of the human reference genome (hg19) and 92 ERCC spike-in control sequences, was used for mapping. Using the htseq-count Python code[57], the read counts per mRNA were estimated. They were then normalized to the count-per-million (CPM) value using EdgeR[58]. The CPM was converted to attomole through linear fitting of the ERCC RNA spike-in. The half-life of the transcript was determined according to a known formula with normalized values[59], and the negative half-life value was ignored. The distribution of the half-life was transformed to a CDF to determine the statistical significance of the difference among groups using Python codes. The two-tailed Mann–Whitney $U$ test was used to determine $p$ values between targets and non-targets.

**Gene ontology**. For GO analysis, the official symbols of genes, which harbored LC3B peaks in the "AAUAAA" motif and showed a decrease in abundance by at least 1.3-fold upon Rapa treatment and an increase in abundance by least 1.3-fold upon LC3B downregulation, were uploaded to DAVID (https://david.ncifcrf.gov/) and then analyzed using the functional annotation tool. The identified GO terms were retrieved from the result files according to biological process terms (level 3, $p$ values $<10^{-2}$) or molecular function terms (level 3, $p$ values $<10^{-2}$).

**mRNA secondary structure analysis**. A sliding window of 30 nucleotides with a step of 1 nucleotide was used to measure the MFE along each mRNA. For each window, the MFE was calculated using the fold module of the ViennaRna package 2.0[60] with default parameters. GC content was also calculated in the same window. The MFE of sequences between the termination codon and PAS was measured using the fold_compound module of ViennaRna package 2.0. The two-tailed Kolmogorov–Smirnov test was conducted to determine the $p$ values of the differences among the groups for MFE and GC content analysis.

**Sequence alignment**. Alignment of multiple amino acid sequences of LC3B among various species was performed by CLUSTALW from BioEdit v7.2.5[61] and visualized by ESPript3[62].

**Quantitative real-time RT-PCR**. After cell harvesting, total RNA was purified with TRIzol Reagent (Life Technologies) and synthesized into cDNA using RevertAid Reverse Transcriptase (Thermo Scientific). qRT-PCR analyses were performed with gene-specific primers and the Light Cycler 480 SYBR Green I Master Mix (Roche) on a Light Cycler 480 II machine (Roche). The oligonucleotides used in this study are listed in Supplementary Data 3b.

**Immunoprecipitation**. HEK293T cells were harvested and lysed using NET-2 buffer [50 mM Tris-HCl (pH 7.4), 150 mM NaCl, 1 mM phenylmethylsulfonyl fluoride (PMSF; Sigma-Aldrich), 2 mM benzamidine hydrochloride (Sigma-Aldrich), 0.05% NP-40 (IGEPAL® CA-630; Sigma-Aldrich), 10 mM sodium

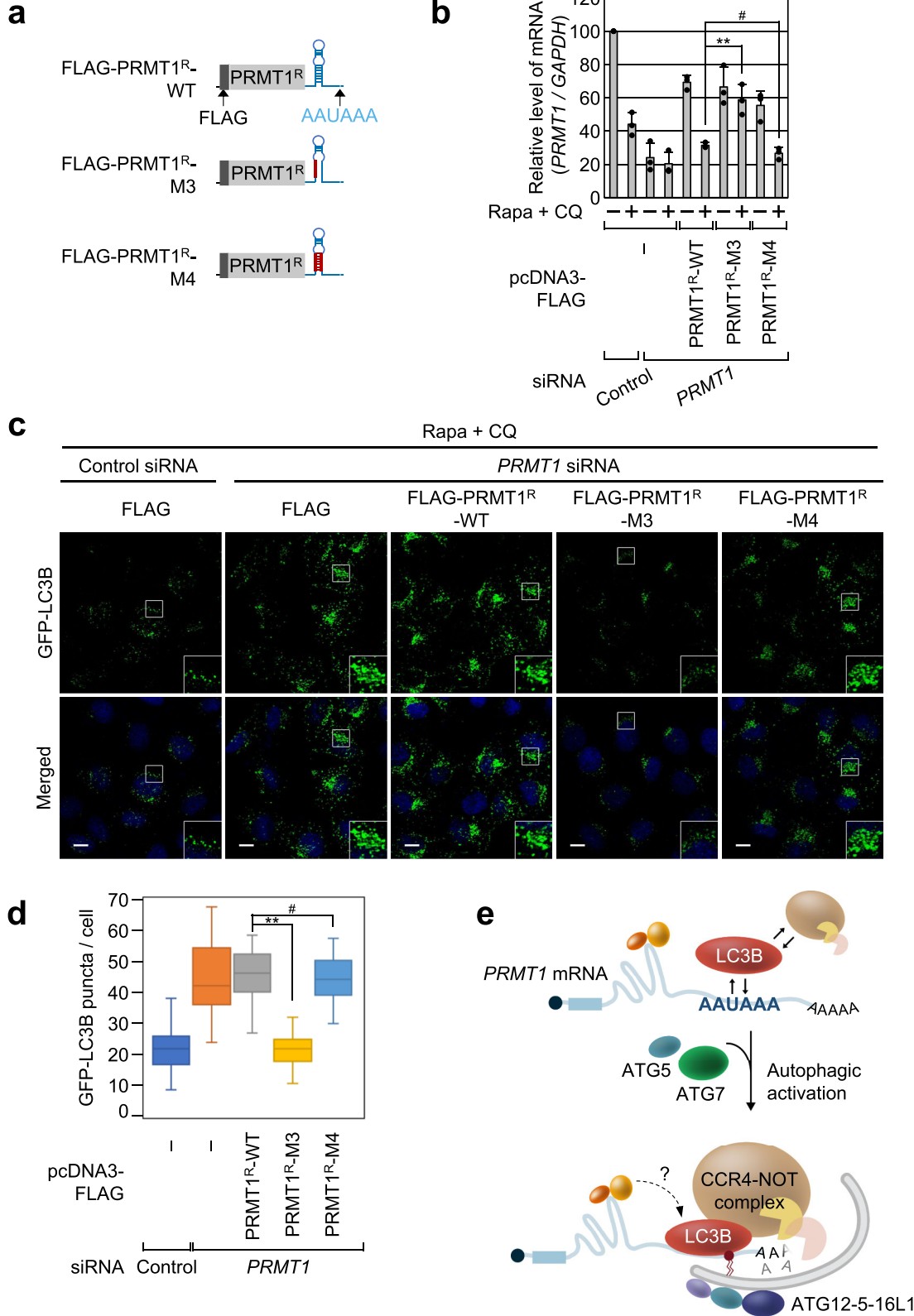

fluoride (Sigma-Aldrich), and 0.25 mM sodium orthovanadate (Sigma-Aldrich)]. The cell extracts were sonicated and precleared with protein G agarose 4B beads (Incospharm) for 1 h at 4 °C. Next, the samples were incubated with either antibody-conjugated beads or FLAG M2 affinity gel (Sigma-Aldrich) at 4 °C. After 3 h, the beads were washed four times with NET-2 buffer and eluted with 2× sample buffer. The samples were analyzed using western blotting.

**In vivo CLIP**. In vivo CLIP was performed as CLIP-seq with slight modifications, as described previously[63]. HEK293T cells were washed with ice-cold PBS and irradiated with 254 nm and 400 mJ/cm² UV light using a UV cross-linker before harvesting. The cross-linked cell pellets were mixed with 600 μl of CLIP-seq lysis buffer and incubated for 10 min on ice. The cell lysates were treated with RNase-free DNase I (30 U) and incubated at 37 °C for 5 min. After centrifugation at

**Fig. 8 Rapid degradation of the *PRMT1* mRNA via LMD promotes autophagy. a** Schematic diagram of siRNA-resistant (R) FLAG-PRMT1[R] reporters, WT, M3, or M4 mRNA. These reporter mRNAs are the same as RLuc-P3'-WT, M3, and M4 mRNA described in Fig. 4c, except that the FLAG-PRMT1[R] reporter mRNAs encode the full-length PRMT1 protein instead of RLuc. **b–d** Complementation experiment using the FLAG-PRMT1[R] reporter mRNAs. HeLa cells stably expressing GFP-LC3B were transiently transfected with either *PRMT1* siRNA or nonspecific Control siRNA. Two days later, the cells were retransfected with a plasmid expressing either FLAG or one of the FLAG-PRMT1[R] reporter mRNAs. The cells were treated with Rapa + CQ for 12 h before immunostaining with the α-GFP antibody. *n* = 3. **b** Relative level of *PRMT1* mRNAs. *n* = 3. Data are presented as mean values ± SD; # not significant; **$p < 0.01$ (The exact *p* values are provided in Source Data file). **c** Immunostaining of GFP-LC3B. Scale bar = 10 μm. **d** Quantitation of GFP-LC3B puncta per cell. Box-whiskers show maximum, third quartile to first quartile, median and minimum. A one-way ANOVA test was conducted to calculate the *p* values; *n* = 865 cells examined over three independent experiments; # not significant; **$p < 0.01$. **e** Proposed model illustrating the role of LMD in autophagy. Source data are provided as a Source Data file.

---

13,000 × *g* for 20 min at 4 °C, the supernatants were mixed with a primary antibody preincubated with protein A Dynabeads for 2 h at 4 °C and subjected to IP. The beads were washed twice with CLIP-seq lysis buffer and washed once with CLIP-seq PNK buffer. The resin-bound protein and RNA were analyzed by western blotting and qRT-PCR, respectively.

**Immunostaining**. HeLa cells or HeLa cells stably expressing GFP-LC3B were fixed with 3.8% formaldehyde (Sigma-Aldrich) for 30 min and permeabilized with 0.5% Triton X-100 (Sigma-Aldrich) for 10 min. The cells were incubated with 1.5% BSA (BovoStar) for blocking. After incubation for 1 h, the cells were incubated with a primary antibody diluted in 0.5% BSA and then incubated with the secondary antibody diluted in 0.5% BSA for 1 h. The nuclei were stained with 4′,6-diamidino-2-phenylindole (DAPI; Biotium). Immunostained signals were analyzed using Zeiss LSM 510 Meta, Zeiss LSM 700, or Zeiss LSM 800 confocal microscopy.

**In situ proximity ligation assay**. In situ PLA was performed using a PLA kit (Sigma-Aldrich) according to the manufacturer's instructions. In brief, cells were washed twice with ice-cold PBS and fixed in 3.65–3.8% formaldehyde for 30 min at room temperature. The cells were then permeabilized with 0.5% Triton X-100 for 10 min at room temperature. After washing three times with PBS, the cells were incubated in blocking buffer (Sigma-Aldrich) for 1 h at 37 °C. After blocking, the cells were incubated with the primary antibody diluted in antibody diluent (Sigma-Aldrich) for 1 h at 37 °C. The cells were then incubated with PLUS or MINUS PLA probes (Sigma-Aldrich) for 1 h at 37 °C. Ligation was performed for 30 min at 37 °C using a ligase specific for PLA (Sigma-Aldrich). After ligation, the cells were incubated in a polymerase solution containing a polymerase diluted in Duolink Amplification Green (Sigma-Aldrich) for 100 min at 37 °C. DAPI was used to stain the nuclei, and images were obtained using a Zeiss LSM 800 confocal microscope.

Where indicated, the PLA experiments were coupled to immunostaining experiments. After incubation of the cells in a polymerase solution during the PLA experiment, the cells were incubated with primary and secondary antibodies for immunostaining. The nuclei were stained with DAPI, and images were visualized using a Zeiss LSM 800 confocal microscope.

**Nucleocytoplasmic fractionation**. HEK293T cells were harvested in ice-cold PBS and subjected to centrifugation. The cell pellets were mixed with hypotonic buffer [10 mM Tris-HCl (pH 7.4), 10 mM NaCl, 0.1% Triton X-100, 10 mM EDTA, 1 mM PMSF, and 2 mM benzamidine hydrochloride] for 10 min on ice. Centrifugation was performed at 13,000 × *g* for 10 min at 4 °C. The supernatant was collected for the cytoplasmic fraction. The pellets were washed three times with NET-2 buffer and resuspended in hypotonic buffer for sonication. Sonication was performed by 30 bursts of 1 s each (3 output control, 30% duty cycle, Sonifier 250, Branson). After sonication, the lysates were centrifuged at 13,000 × *g* for 15 min at 4 °C. The supernatant was collected for the nuclear fraction. Proper fractionation was confirmed using western blotting with antibodies against U1 snRNP70 (a nuclear protein) and GAPDH (a cytoplasmic protein).

**LC3B purification**. Mature LC3B WT (1–120) or its R/Q variant was expressed as the C-terminal MBP-fused form in *Escherichia coli* BL21(DE3). After induction with 1.0 mM IPTG at 0.7. OD$_{600}$, the cells were grown for 20 h at 18 °C. The MBP-fused LC3B, either WT or R/Q variant, was purified by amylose affinity column chromatography, and then, the MBP was cleaved by human ATG4B[64]. Further purification was performed using an anion exchange column, HiTrap SP Fast Flow (GE Healthcare, 17-5054-01). The eluted proteins were loaded onto a HiLoad$^{TM}$ 16/600 Superdex$^{TM}$ 75 pg (GE Healthcare, 28-9893-33) gel filtration column equilibrated with a buffer containing 40 mM Tris-HCl (pH 8.0), 30 mM KCl, 1 mM MgCl$_2$, and 1 mM DTT. All buffers used for purification were prepared using DEPC-treated water.

**Electrophoretic mobility shift assay**. A Cy5-labeled single-stranded RNA probe harboring three tandem repeats of the LC3B binding consensus motif (Cy5-AAUAAAAAUAAAAAUAAA, 2 μM) or the corresponding negative control RNA

(Cy5-AAAAAAAAAAAAAAAAAA, 2 μM) was incubated with purified BSA, LC3B, or LC3B-R/Q at 30 °C for 1 h in EMSA binding buffer [40 mM Tris (pH 8.0), 30 mM KCl, 1 mM MgCl$_2$, 0.01% NP-40, and 1 mM DTT]. After incubation, the resulting mixtures were subjected to electrophoresis using a non-denaturing 8% polyacrylamide gel. The gel was exposed and imaged using the Odyssey Imaging System (Li-COR).

**Fluorescence polarization assay (FP assay)**. Cy5-labeled RNAs were dissolved to 200 nM concentration in binding buffer [40 mM Tris-HCl (pH 8.0), 30 mM KCl, 1 mM MgCl$_2$, and 1 mM DTT] and aliquoted with 15 μl volume to reaction well. Purified LC3B-WT or LC3B-R/Q was serially diluted in binding buffer and added to each reaction well to final volume 30 μl at a concentration ranging from 3.4 to 880 μM. To detect the change in light polarization of the Cy5-labeled RNA, fluorescent measurements were performed in a 384-well format on a Corning black low-volume plate using SpectraMax® ID5 (Molecular Device) with excitation and emission wavelengths of 640 and 682 nm, respectively. A nonlinear graph of LC3B concentration-dependent polarization was calculated and drawn using GraphPad Prism 9.

**Poly(A) tail-length assay**. Poly(A) tail-length assay was performed using a Poly(A) tail-length assay kit (Thermo Scientific) according to the manufacturer's instructions. In brief, total RNA was purified using TRIzol Reagent from HEK293T cells and treated with RNase-free DNase I to remove DNA. The G/I tailing mix was incubated with 2 μg RNA for 1 h at 37 °C. After incubation for 1 h, a tail stop solution was added. The RNA was then incubated with RT mix at 44 °C for 1 h and 92 °C for 10 min to synthesize cDNA. The cDNAs were amplified by PCR using PCR mix, DNA polymerase, and PCR primers. The specific oligonucleotides used in this study are listed in Supplementary Data 3c. The PCR products were loaded onto an agarose gel and stained with ethidium bromide.

**Statistical analysis**. Data in bar graphs, mRNA half-life analysis (Fig. 3e–g), and quantification of coimmunoprecipitated proteins were analyzed using two-tailed and equal-variance Student's *t* test to define *p* values < 0.05 or < 0.01. All data were obtained from at least three independent biological replicates and are presented as the mean ± standard deviation.

The two-tailed Kolmogorov–Smirnov test was used for statistical analysis to determine the distance between LC3B peaks located at the consensus AAUAAA motif and PAS. The two-tailed Kolmogorov–Smirnov test was also conducted to determine *p* values for secondary structure analysis. In addition, the Wilcoxon signed-rank test and two-tailed Mann–Whitney *U* test were performed for the heat map and CDF analysis, respectively.

To determine the number of GFP-LC3B puncta or PLA spots per cell, at least 50 individual cells from each biological replicate were visualized using LSM 510 Meta or LSM 800. The number of GFP-LC3B puncta and PLA spots was measured using ImageJ software. For statistical comparisons, one-way ANOVA followed by Tukey's honestly significant difference test was performed with significance defined as a *p* value <0.01.

**Reporting summary**. Further information on research design is available in the Nature Research Reporting Summary linked to this article.

## Data availability

The accession number for the next-generation sequencing data reported in this paper is the NCBI Sequence Read Archive (SRP303078). RefSeq Genes were obtained from the UCSC table browser hg19 (http://genome.ucsc.edu/cgi-bin/hgTables). Source data are provided with this paper.

## Code availability

In-house shell scripts and python and R codes used for analyzing our NGS data have been deposited in Zenodo database under accession code https://doi.org/10.5281/zenodo.5968564.

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

## Acknowledgements

We thank Dr. Chungho Kim for providing HeLa cells stably expressing GFP-LC3B. This work was supported by a National Research Foundation (NRF) of Korea grant funded by the Korean government (Ministry of Science, ICT, and Future Planning; NRF-2015R1A3A2033665 and NRF-2018R1A5A1024261). H.H. was supported in part by the Basic Science Research Program through the NRF, funded by the Ministry of Education (NRF-2019R1I1A1A01058792).

## Author contributions

H.J.H., B.S.L., and Y.K.K. designed the experiments. H.J.H. and B.S.L. performed the experiments and analyzed the data. H.H. analyzed the sequencing data. B.H.K. purified recombinant LC3B proteins under the supervision of H.K.S. H.J.H., B.S.L., and Y.K.K. wrote the manuscript. Y.K.K. supervised the study.

## Competing interests

The authors declare no competing interests.
