## [Peer Review File · Nature Communications]

REVIEWER COMMENTS

Reviewer #1 (Remarks to the Author):

Summary:

In this manuscript the authors describe a mechanism by which autophagy and mRNA stability pathways are linked. The process is termed LC3B-mediated decay (LMD). Using biochemical and cell-based approaches LC3B is shown to interact with 3'UTRs via the polyadenylation sequence, AAUAAA. Upon autophagic induction and LC3B cleavage, LC3B binds specific AAUAAA-containing mRNAs, recruits CCR4-NOT complex and triggers degradation, presumably via deadenylation. Overall this is a very exciting manuscript with important implications for mRNA degradation in the context of autophagy. However, concerns over direct LC3B binding and the precise mechanism of mRNA degradation should be addressed.

Major Comments:

Motif-based analysis should be repeated by comparing motifs in peaks relative to matched regions without peaks. AU-rich motifs are likely to be found in 3UTRs as most UTRs display enrichments for these sequences. Such an analysis may help identify other modes of binding that are not mediated by AAUAAA in 3UTRs and transcript regions.

Little attention is given to peaks outside of 3UTRs, can they authors discuss what happens to targets with binding sites and motifs outside of the 3UTR (e.g mRNA levels, half-life)?

It would be helpful if the authors provided additional CLIP analysis such as reads/fpkms per peak with and without motif, PCR duplicates, cross-replicate correlations, and peak overlap.

While the EMSA presented indeed shows a shift it is minimal relative to free RNA. It would be helpful to further assess direct binding. Since the authors seek to establish LC3B as an RBP, measuring affinity by filter assay, SPR, fluorescence polarization, OR ITC would help strengthen the claims of direct RNA binding. More robust biophysical assays will be important when cross-comparing LC3B binding to different motifs.

Direct RNA binding of LC3B to a natural target containing the structural features discussed below would strengthen the claim that LC3B is an RNA binding protein.

“We chose these residues within LC3B because the arginine-rich motif is one of the common features found in many RBPs” While this is true, it remains unclear if disordered R-rich or SR-rich motifs within RBPs mediate direct and sequence-specific binding. It appears the mutated R residues in LC3B lie within a well-structured helix rather than a disordered region. The authors should expand on their selection for these mutants. It would be helpful if these residues could be mapped onto the structure of LC3B. Species conservation of these residues should be presented.

“Collectively, our CLIP-seq data and EMSA data indicate that LC3B directly binds to RNA with a high preference for the AAUAAA motif.” This statement is mainly supported by assessing 3 other short motifs in vitro with EMSAs that are not quantified. Additionally, many CLIP peaks were not at the AAUAAA motif.

“suggesting that efficient LMD requires a conversion of LC3B.” Does cleavage impact direct RNA binding by LC3B? This can be tested in vitro.

In comparing half-lives, the authors demonstrate that downregulation of LC3B increases half-life of CLIP targets in cells treated with RAPA+CQ. This is an intriguing find suggesting that LMD, to some extent, is always downregulating targets. Can the authors show half-life measurements for siLC3B/siCTRL in DMSO conditions? Are targets substantially stabilized? To what extent is LC3B acting on mRNAs in conditions where autophagy is not synthetically activated or suppressed.

In figure 2F, how was the long-range structure determined? Can data to support these structures be shown? Perhaps quantitation of the fraction of targets with this structural configuration? Conversely, since the structure does not appear to be impact LMD perhaps this can be omitted.

In the analysis of RNA structure of targets, can the authors clarify how non-targets were selected for MFE determination? As shown it seems most expressed mRNAs are binned into non-targets. It might be important to select a control group of non-targets with matched distances between the AAUAAA and the STOP codon. Generally, the 3'UTR and CDS have different GC content which may impact overall structure determination between the two groups.

“The role of the stable hairpin structure in LMD was further validated through in vivo CLIP experiments using α -LC3B antibody and qRT-PCRs” Are the authors proposing that a stable stem is required for direct LC3B binding or for overall LMD activity? If direct binding, this could be tested in vitro.

The authors demonstrate that LMD likely requires CCR4-NOT but deadenylation is never directly shown nor is assembly on a transcript in vivo or in vitro. Demonstrating LC3B-mediated recruitment of CCR4-NOT and subsequent deadenylation will further establish LMD as a key degradation pathway.

Minor comments:

Why is the distal 1/5 of UTRs selected for motif analysis? What proportion of peaks lie in this region?

The y-axis in Fig. 2E seems a bit strange or non-linear, can the authors explain?

The section: “LC3B elicits a rapid decay of LMD substrates via a CCR4-NOT complex” could use more explanation on the PLA assay used.

Have efforts to identify RNA bound proteins by the Hentze and Landthaler labs identified LC3B as an RNA binding protein?

More discussion on the mode of RNA binding should be included. In looking at the LC3B structure and the mutated residues it is not immediately obvious how this protein might bind RNA in such a specific manner.

Reviewer #2 (Remarks to the Author):

In this interesting study, Hwang et al report a role for LC3B in autophagy regulation via RNA binding. The authors discover an LC3B RNA binding consensus sequence which together with mRNA secondary structure drives specificity of LC3B recognition. LC3B binding to the mRNA of PRMT1 was found to be important for the turnover of the mRNA (via CCR4-NOT) and was necessary for efficient autophagy (as measured by LC3B foci). Curiously, although lysosome activity was not required for the degradation of mRNA, LC3B lipidation was found to be necessary. Overall, this study will appeal to a wide audience and will be of great interest to researchers in the autophagy field. The manuscript is well written and clear, and the experimental data are clear and convincing, and are mostly well controlled. There are some areas where additional controls and additional experiments would help strengthen the authors conclusions and provide more mechanistic insight into the RNA binding role of LC3B in autophagy.

Specific comments:

1. The study relies on mTOR inhibition via Rapamycin. Can the authors observe the same results with respect to PRMT1 mRNA degradation during starvation induced autophagy? This would be important to address in order to clarify whether this is a general mechanism of autophagy regulation that isn't specific only to Rapamycin treatment.
2. Figure 3B: The authors show that PRMT1 mRNA levels are decreased and that this is dependent on LC3B mRNA binding but have not shown whether this has any effect on PRMT1 protein levels. This is an important control to include (using Rapa +CQ, and under starvation conditions).
3. Extended Data S9: It would be more convincing if the can the authors used an endogenous marker of autophagic puncta instead (e.g. WIPI2), rather than over expressing LC3B.
4. Figure 4B-D: Can the authors show the expression levels of the different FLAG-PRMT1 constructs before and after Rapa +CQ treatment to ensure that they are equally expressed and that the protein levels are affected by LC3B mediated mRNA degradation?
5. Figure 4D: In addition to looking at LC3B puncta, can the authors also analyse autophagy substrate turnover by conducting the same experiment (but minus CQ), and also under starvation conditions, combined with western blotting of p62 to assess its autophagic turnover. This is important because the formation of LC3B foci does not necessarily correlate with lysosomal turnover of autophagy substrates.
6. Is LC3B lipidation necessary for binding to CNOT1 and CNOT7? This would provide some mechanistic insight into the degradation mechanism. Given that LC3B binds CCR4-NOT predominantly outside of

autophagosome puncta, it is possible that LC3B will bind CCR4-NOT in the absence of lipidation, but that the partial colocalization of this complex on autophagosomes is necessary for its mRNA degrading capability (and potentially the site of degradation). This experiment could be conducted in ATG5 siRNA cells +/- Rapamycin treatment.

Reviewer #3 (Remarks to the Author):

The manuscript submitted by Hwang, et al., presents a series of well-executed experiments that definitively show that LC3B is an RNA binding protein that specifically recognizes and interacts with a consensus motif (AAUAAA + secondary structure upstream) located at the 3' UTR of target transcripts. The authors further demonstrate that this RNA-binding functionality of LC3B funnels the mRNA cargo for degradation to the CCR4-NOT complex when autophagy is triggered by the addition of rapamycin. Furthermore, the authors show that targeted degradation of one of its target mRNAs, PRMT1, a known negative regulator of autophagy, promotes autophagy.

Overall, the experiments presented in this paper are well thought out and well executed. However, there were a few aspects in the paper that could have been further explored.

1. On page 10, the authors ask, "when and where does LC3B bind to LMD substrates within the cells?" The authors show through fractionation experiments that LC3B-I and LC3B-II were primarily found in the cytoplasmic fraction. However, multiple studies have shown that LC3 cycles in and out of the nucleus. In 2015, it was shown by Huang, et al., (Mol Cell) that LC3 shuttles between the nucleus and the cytoplasm through the action of the SIRT1 deacetylase. I am curious to see how mutations in LC3B that prevent their shuttling from the nucleus to the cytoplasm affect substrate binding and subsequent LMD.
2. Depletion of LC3B is not very efficient (Fig. S2a). Could a more efficient knockdown of LC3B affect the abundance of other identified cargo?
3. It is interesting to see that the mRNA interaction with LC3B requires its ability to conjugate with PE (Fig. 3c), and mRNA, LC3B-II, and the CCR4-NOT1 complex are recruited to the phagophore (Fig. S9 and 10 and Fig. 4e) but mRNA is not degraded through autophagy. I think there could be more discussion about why the interaction between LC3B-II and mRNA is not enough to degrade the mRNA through autophagy (mRNA as a cargo) but needs the help of other proteins.
4. Is the LMD inhibited if there no interaction between LC3B and the CCR4-NOT1 complex? What is the mRNA or protein level of PRMT1 in the CNOT1 or CNOT7 knockdown or knockout cells? Also, what about the autophagy activity?
5. Does the binding between LC3B and CNOT1 or CNOT7 need LC3B lipidation? Is the increasing binding after rapamycin treatment (Fig. 3d) due to the higher affinity to LC3B-II? Can these two proteins bind to LC3B that lacks the ability to conjugate with PE, such as LC3BG120A?

6. Besides PRMT1, there are some other genes with a big change in mRNA level before and after knocking down LC3 as seen in Fig. S4, such as DDIT4, which has been shown to regulate mTOR (Foltyn et al, 2019), so that it may have a role in autophagy regulation. Are there any other ways that LMD regulates autophagy?

7. The sections involving analyses with regards to CCR4-NOT interaction could be better written. For example, the analysis of experiments found in Extended Figure 9-10 and parts of Fig. 3 was written in the Discussion section. This could have been included on page 12 when Fig. 3 was initially discussed. A re-write could make this section of the paper flow better.

8. The introduction section also seems to be inadequate, especially when considering that Nature Communications has a broad reach.

** See Nature Research's author and referees' website at www.nature.com/authors for information about policies, services and author benefits.

Responses to reviewer #1's comments

Summary:

In this manuscript the authors describe a mechanism by which autophagy and mRNA stability pathways are linked. The process is termed LC3B-mediated decay (LMD). Using biochemical and cell-based approaches LC3B is shown to interact with 3'UTRs via the polyadenylation sequence, AAUAAA. Upon autophagic induction and LC3B cleavage, LC3B binds specific AAUAAA-containing mRNAs, recruits CCR4-NOT complex and triggers degradation, presumably via deadenylation. Overall this is a very exciting manuscript with important implications for mRNA degradation in the context of autophagy. However, concerns over direct LC3B binding and the precise mechanism of mRNA degradation should be addressed.

- **We appreciate the reviewer's interest in our study and valuable comments. Kindly go through our detailed responses below.**

Major Comments:

Motif-based analysis should be repeated by comparing motifs in peaks relative to matched regions without peaks. AU-rich motifs are likely to be found in 3UTRs as most UTRs display enrichments for these sequences. Such an analysis may help identify other modes of binding that are not mediated by AAUAAA in 3UTRs and transcript regions.

- **We appreciate the reviewer's suggestion. We repeated the similar motif analysis using sequences in the 3'UTR without peaks. The results showed that motifs in the 3'UTR without peaks did not include AAUAAA. Therefore, we would not like to include these results in the revised manuscript. Instead, the new analyses are presented in Figure R1 (reviewer only).**

Little attention is given to peaks outside of 3UTRs, can they authors discuss what happens to targets with binding sites and motifs outside of the 3UTR (e.g mRNA levels, half-life)?

- **We apologize for this poor description. In the original manuscript, we analyzed the role of LC3B-binding in mRNA abundance and half-life in Fig. 1f-i (Fig. 2 in the revised manuscript). A cumulative distribution function (CDF) analysis showed that the amounts and half-lives of mRNAs were more significantly**

reduced when LC3B was bound to the 3'UTR, relative to the 5'UTR or coding sequence. For a better description and presentation, we have reorganized the order of the figures and all transcriptomic data related to the LC3B-binding effect on mRNA abundance and half-life were moved to Fig. 2 in the revised manuscript.

It would be helpful if the authors provided additional CLIP analysis such as reads/fpkms per peak with and without motif, PCR duplicates, cross-replicate correlations, and peak overlap.

- **As suggested, we additionally analyzed the CLIP-seq data during revision and observed the following: (i) 3'UTR_AAUAAA peaks showed a comparable level of reads/FPKM to 3'UTR_No_AAUAAA peaks, suggesting that LC3B binds to AAUAAA motif and other uncharacterized motif(s) in the 3'UTR with comparable levels; (ii) we reanalyzed CLIP-seq data with or without PCR duplicates and obtained the same metagene distributions and AAUAAA motif regardless of PCR duplicates. Therefore, we would like to retain the original metagene data (Fig. 1b). Instead, the metagene analysis and motif analysis with CLIP-seq data after removing PCR duplicates using Picard (MarkDuplicates) are presented in Figure R2, and (iii) our original manuscript included all information on the correlation between two biological replicates of CLIP-seq (Supplementary Table 1).**

While the EMSA presented indeed shows a shift it is minimal relative to free RNA. It would be helpful to further assess direct binding. Since the authors seek to establish LC3B as an RBP, measuring affinity by filter assay, SPR, fluorescence polarization, OR ITC would help strengthen the claims of direct RNA binding. More robust biophysical assays will be important when cross-comparing LC3B binding to different motifs.

- **We appreciate the reviewer's comments. To demonstrate that LC3B is an RNA-binding protein, we carried out several experiments, such as EMSA, CLIP-seq, and *in vivo* CLIP, all of which allow for monitoring direct interactions between RNA and protein. Notably, CLIP-seq and *in vivo* CLIP experiments are popularly used to demonstrate the direct interaction between**

RNA and proteins within cells. In our manuscript, all data obtained from the three independent approaches consistently support that LC3B directly interacts with RNA.

- **To ease the reviewer’s concerns, however, we performed fluorescence polarization assay with purified proteins (LC3B-WT and LC3B-R/Q) and Cy5-labeled probes (Cy5-ACGCCAAAAAAA and Cy5-ACGCCAAUAAAA). The results supported preferential interaction between LC3B and AAUAAA motif. These new data have been added to Fig. 1f in the revised manuscript.**

Direct RNA binding of LC3B to a natural target containing the structural features discussed below would strengthen the claim that LC3B is an RNA binding protein.

“We chose these residues within LC3B because the arginine-rich motif is one of the common features found in many RBPs” While this is true, it remains unclear if disordered R-rich or SR-rich motifs within RBPs mediate direct and sequence-specific binding. It appears the mutated R residues in LC3B lie within a well-structure helix rather than a disordered region. The authors should expand on their selection for these mutants. It would be helpful if these residues could be mapped onto the structure of LC3B. Species conservation of these residues should be presented.

- **First of all, we would like to mention that many SR-rich motifs within RBPs are not disordered. They lie within a well-structured helix rather than a disordered region.**
- **To address the reviewer’s comments, we carried out several experiments as follows: First, we performed *in vivo* CLIP experiments, which allowed us to monitor a direct interaction between RNA and protein within cells, using either LC3B-WT or R/Q. The *in vivo* CLIP results showed that LC3B-WT, but not the LC3B-R/Q mutant, directly binds to endogenous target mRNAs (*PRMT1* mRNAs and *MARS1* mRNAs). These new data have been added to Supplementary Fig. 6 in the revised manuscript.**
- **Second, we analyzed the known structure of LC3B. The RRR motif was located on the surface of LC3B. This structural image has been added to Supplementary Fig. 2b in the revised manuscript.**
- **Finally, as commented, we analyzed the possible conservation of the RRR**

motif in LC3B among various species. Interestingly, the RRR motif was highly conserved among the tested species. The data on sequence comparison have been added to Supplementary Fig. 2c in the revised manuscript.

“Collectively, our CLIP-seq data and EMSA data indicate that LC3B directly binds to RNA with a high preference for the AAUAAA motif.” This statement is mainly supported by assessing 3 other short motifs *in vitro* with EMSAs that are not quantified. Additionally, many CLIP peaks were not at the AAUAAA motif.

- **We would like to emphasize that CLIP-seq allows for direct interaction between RNA and proteins. Using this approach, we identified the AAUAAA motif as one of the LC3B-binding sites. To validate these observations more clearly, we carried out EMSA using a probe with the AAUAAA motif. In addition to EMSA, we also performed *in vivo* CLIP experiments using reporter RLuc mRNAs containing either the wild-type (AAUAAA) motif or a single nucleotide substitution (AAUAAA to AAAAAA) of *PRMT1* 3'UTR. Our *in vivo* CLIP experiments showed that LC3B preferentially interacts with *PRMT1* WT than the *PRMT1* mutant. These data support that LC3B directly binds to the AAUAAA motif.**
- **As mentioned, some CLIP peaks were not present in the AAUAAA motif. This comment was also related to the reviewer's first major comment. We reanalyzed the CLIP-seq data under different conditions during revision. However, the AAUAAA motif in the 3'UTR was the only one with a significant P value. Please refer to Figure R3.**

“suggesting that efficient LMD requires a conversion of LC3B.” Does cleavage impact direct RNA binding by LC3B? This can be tested *in vitro*.

- **In the above sentence, the conversion refers to the conversion of LC3B-I to LC3B-II, but not a conversion of pro-LC3B to LC3B-I. For a clearer description, the sentence was changed as follows: “efficient LMD requires a conversion of LC3B-I to LC3B-II”**

In comparing half-lives, the authors demonstrate that downregulation of LC3B increases

half-life of CLIP targets in cells treated with RAPA+CQ. This is an intriguing find suggesting that LMD, to some extent, is always downregulating targets. Can the authors show half-life measurements for siLC3B/siCTRL in DMSO conditions? Are targets substantially stabilized? To what extent is LC3B acting on mRNAs in conditions where autophagy is not synthetically activated or suppressed.

- **This is an intriguing question. In the present study, we focused on LMD during autophagic induction. However, LC3B might be involved in mRNA stability, regardless of autophagy under normal conditions. During revision, we reanalyzed transcriptome data and observed that the half-life of the LC3B CLIP group was reduced upon LC3B downregulation under normal conditions, suggesting that LC3B functions as a stabilizer under normal conditions. Although these observations are intriguing, we focused on LC3B-mediated mRNA degradation under autophagic conditions in this study. Instead, we would like to present the data in Figure R4 only for the reviewer.**

In figure 2F, how was the long-range structure determined? Can data to support these structures be shown? Perhaps quantitation of the fraction of targets with this structural configuration? Conversely, since the structure does not appear to be impact LMD perhaps this can be omitted.

- **We apologize for the poor description. We did not determine the long-range structure shown in Fig. 2f in the original manuscript (Fig. 4a in the revised manuscript). Instead, we found from bioinformatic analysis that mRNAs harboring the 3'UTR AAUAAA motif that binds to LC3B (3'UTR_AAUAAA group) tended to form a long-range base-pairing between the region immediately downstream of the AAUAAA motif and the region immediately downstream of a translation termination codon. However, long-range base-pairing was not essential for efficient LMD (Supplementary Fig. 7c in the revised manuscript). To describe our observations more clearly, we have rearranged the Figures and added them to Fig. 4 and Supplementary Fig. 7 in the revised manuscript.**

In the analysis of RNA structure of targets, can the authors clarify how non-targets were

selected for MFE determination? As shown it seems most expressed mRNAs are binned into non-targets. It might be important to select a control group of non-targets with matched distances between the AAUAAA and the STOP codon. Generally, the 3'UTR and CDS have different GC content which may impact overall structure determination between the two groups.

- **We analyzed the RNA structure based on the MFE value, as described in the Materials and Methods section. In Fig. 2g (now Fig. 4b), we compared the MFE values of the 3'UTR sequences of the 3'UTR_AAUAAA group and those of nontarget mRNAs (all other mRNAs). The details are described in the Figure Legends and Materials and Methods section of the revised manuscript.**

“The role of the stable hairpin structure in LMD was further validated through *in vivo* CLIP experiments using α -LC3B antibody and qRT-PCRs” Are the authors proposing that a stable stem is required for direct LC3B binding or for overall LMD activity? If direct binding, this could be tested *in vitro*.

- **All experiments suggested by the reviewer have been included in the original manuscript. First, LC3B binding to RNA was promoted by the structure (Fig. 4d in the revised manuscript). Second, the increased binding of LC3B to mRNA by the structure increased the LMD efficiency (Fig. 4e in the revised manuscript).**
- **As mentioned above, our *in vivo* CLIP experiment allowed us to determine the direct interaction between RNA and protein within the cells. Although we did not perform *in vitro* experiments, the current data strongly support that a direct interaction between LC3B and target mRNA is promoted by the presence of a structure upstream of the AAUAAA motif.**

The authors demonstrate that LMD likely requires CCR4-NOT but deadenylation is never directly shown nor is assembly on a transcript *in vivo* or *in vitro*. Demonstrating LC3B-mediated recruitment of CCR4-NOT and subsequent deadenylation will further establish LMD as a key degradation pathway.

- **As suggested, we assessed the change in the length of poly(A) of endogenous LMD substrates (*COTL1* mRNA and *CCT7* mRNA). Rapa treatment**

drastically shortened the length of the endogenous LMD substrates. In contrast, LC3B downregulation caused inefficient shortening under Rapamycin-treated conditions. Therefore, these data support the active role of the CCR4-NOT complex in LMD. Of note, we could not compare the relative change in the length of endogenous *PRMT1* mRNA because of its very short poly(A) tail under our conditions. All new data have been added to Supplementary Fig. 11 in the revised manuscript.

Minor comments:

Why is the distal 1/5 of UTRs selected for motif analysis? What proportion of peaks lie in this region?

- **From the metagene analysis, we observed that the distal 1/5 of the 3'UTR exhibited drastic enrichment of LC3B binding. Because of this property, we analyzed the distal 1/5 of the 3'UTR. As commented by the reviewer, we also think that this is arbitrary. Therefore, during revision, we reanalyzed motifs using LC3B-binding sites throughout the entire 3'UTR rather than the distal 1/5 of the 3'UTR. The original data were replaced with new data in the revised manuscript (Fig. 1c).**

The y-axis in Fig. 2E seems a bit strange or non-linear, can the authors explain?

- **In general, the half-life analysis is presented as logarithmic scale (log2). Fig. 2e was moved to Fig. 3e in the revised manuscript.**

The section: “LC3B elicits a rapid decay of LMD substrates via a CCR4-NOT complex” could use more explanation on the PLA assay used.

- **For a better description on PLA assay, we slightly modified the sentence. In addition, a previous paper on PLA was cited in the revised manuscript.**

Have efforts to identify RNA bound proteins by the Hentze and Landthaler labs identified LC3B as an RNA binding protein?

- **We examined the RNA-binding proteins characterized by Hentze lab (Castello**

et al., 2012, Cell) and Landthaler lab (Baltz et al., 2012, Molecular Cell). LC3B (or ATG8) was not listed as an RNA-binding protein in either study. The papers have also been cited in the revised manuscript.

More discussion on the mode of RNA binding should be included. In looking at the LC3B structure and the mutated residues it is not immediately obvious how this protein might bind RNA in such a specific manner.

- **As mentioned, we briefly discussed LC3B-binding to target mRNAs as follows: “Under normal conditions, LC3B preferentially associates with target mRNAs harboring the AAUAAA motif. The association and specific loading of LC3B onto target mRNAs would be guided by RNA structures and other RBPs upstream of the AAUAAA motif. Upon autophagic activation, LC3B is conjugated to PE. Concomitantly, autophagic activation increases LC3B binding to target mRNAs in a way that is affected by RNA structures and other RBPs upstream of the LC3B-binding site. (Fig. 4). Therefore, it is plausible that LC3B conversion not only causes intracellular redistribution toward the autophagic puncta of LC3B but also reinforces target recognition for LMD.”**
- **According to the above sentences, our model was slightly modified.**

Reviewer #2 (Remarks to the Author)

In this interesting study, Hwang et al report a role for LC3B in autophagy regulation via RNA binding. The authors discover an LC3B RNA binding consensus sequence which together with mRNA secondary structure drives specificity of LC3B recognition. LC3B binding to the mRNA of PRMT1 was found to be important for the turnover of the mRNA (via CCR4-NOT) and was necessary for efficient autophagy (as measured by LC3B foci). Curiously, although lysosome activity was not required for the degradation of mRNA, LC3B lipidation was found to be necessary. Overall, this study will appeal to a wide audience and will be of great interest to researchers in the autophagy field. The manuscript is well written and clear, and the experimental data are clear and convincing, and are mostly well controlled. There are some areas where additional controls and additional experiments would help strengthen the authors conclusions and provide more mechanistic insight into the RNA binding role of LC3B in autophagy.

➤ **We appreciate the reviewer's interest in our study and valuable comments.**

Specific comments:

1. The study relies on mTOR inhibition via Rapamycin. Can the authors observe the same results with respect to PRMT1 mRNA degradation during starvation induced autophagy? This would be important to address in order to clarify whether this is a general mechanism of autophagy regulation that isn't specific only to Rapamycin treatment.

➤ **As suggested, we carried out complementation experiments using *LC3B* siRNA and siRNA-resistant LC3B under serum-starved conditions. The results showed that, as observed in Rapa-treated conditions, serum starvation caused efficient conversion of LC3B and reduction in the abundance of RLuc-P3'-WT reporter mRNA, endogenous LMD substrates, and PRMT1 protein. All these events were reversed by LC3B downregulation. Under these conditions, the expression of siRNA-resistant LC3B-WT, but not LC3B-R/Q, successfully restored LC3B function. These data indicate that LMD is not limited to autophagy induced by Rapa treatment. These new data have been added to Supplementary Fig. 9 in the revised manuscript.**

➤ **In addition, we repeated similar complementation experiments using *PRMT1* siRNA and siRNA-resistant PRMT1 reporters under either serum-starved or Rapa-treated conditions. We observed similar changes in PRMT1 and p62 proteins in the complementation experiment under Rapa-treated conditions or serum-starved conditions. These data also support the evidence that LMD is a general pathway that occurs during autophagic induction. These new data have been added to Supplementary Fig. 13 in the revised manuscript.**

2. Figure 3B: The authors show that PRMT1 mRNA levels are decreased and that this is dependent on LC3B mRNA binding but have not shown whether this has any effect on PRMT1 protein levels. This is an important control to include (using Rapa +CQ, and under starvation conditions).

➤ **As requested, we measured the levels of endogenous PRMT proteins. Consistent with the LMD of *PRMT1* mRNA, the protein levels of PRMT1 were reduced. The new western blotting data have been added to Fig. 5b in the revised manuscript.**

➤ **The second comment is related to comment #1 of the reviewer. As suggested, we analyzed the levels of PRMT1 protein under Rapa-treated conditions or serum-starved conditions. These data have been added to Supplementary Fig. 13 in the revised manuscript. Please also refer to our responses to comment #1 of Reviewer #2.**

3. Extended Data S9: It would be more convincing if the authors used an endogenous marker of autophagic puncta instead (e.g. WIPI2), rather than over expressing LC3B.

➤ **As suggested, we repeated similar PLA experiments using antibody against endogenous p62, another marker of autophagic puncta. In agreement with our conclusions, the PLA spots of LC3B and either CNOT1 or CNOT7 partially overlapped with the autophagic puncta of FLAG-LC3B or endogenous p62. The new data have been added to Fig. 7a and Supplementary Fig. 12 in the revised manuscript.**

4. Figure 4B-D: Can the authors show the expression levels of the different FLAG-PRMT1 constructs before and after Rapa +CQ treatment to ensure that they are equally expressed and that the protein levels are affected by LC3B mediated mRNA degradation?

➤ **To address the reviewer's concerns, we measured the levels of FLAG-PRMT1 protein. The results showed a strong correlation between PRMT1 mRNA and protein levels. The new data have been added to Supplementary Fig. 13a in the revised manuscript.**

5. Figure 4D: In addition to looking at LC3B puncta, can the authors also analyse autophagy substrate turnover by conducting the same experiment (but minus CQ), and also under starvation conditions, combined with western blotting of p62 to assess its autophagic turnover. This is important because the formation of LC3B foci does not necessarily correlate with lysosomal turnover of autophagy substrates.

➤ **To address the reviewer's comments, we repeated similar complementation experiments under either Rapa-treated conditions or serum-starved conditions and assessed the levels of endogenous p62. The results showed a correlation between PRMT1 and p62 protein, suggesting that efficient LMD of *PRMT1* mRNA promotes autophagy. The new data have been added to Supplementary Fig. 13b and c in the revised manuscript.**

6. Is LC3B lipidation necessary for binding to CNOT1 and CNOT7? This would provide some mechanistic insight into the degradation mechanism. Given that LC3B binds CCR4-NOT predominantly outside of autophagosome puncta, it is possible that LC3B will bind CCR4-NOT in the absence of lipidation, but that the partial colocalization of this complex on autophagosomes is necessary for its mRNA degrading capability (and potentially the site of degradation). This experiment could be conducted in ATG5 siRNA cells +/- Rapa CQ treatment.

➤ **We appreciate the reviewer's suggestion. We carried out IP experiments using an α -LC3B antibody and extracts of cells depleted of ATG5 and ATG7. The results showed that treatment with Rapa and CQ increased the association between LC3B and CNOT1 (and CNOT7), which was reversed by downregulation of both ATG5 and ATG7. These data indicate that the**

association between LC3B and CNOT1 (and CNOT7) is promoted by LC3B conversion. These new data have been added to Fig. 6b and c in the revised manuscript.

Reviewer #3 (Remarks to the Author):

The manuscript submitted by Hwang, et al., presents a series of well-executed experiments that definitively show that LC3B is an RNA binding protein that specifically recognizes and interacts with a consensus motif (AAUAAA + secondary structure upstream) located at the 3' UTR of target transcripts. The authors further demonstrate that this RNA-binding functionality of LC3B funnels the mRNA cargo for degradation to the CCR4-NOT complex when autophagy is triggered by the addition of rapamycin. Furthermore, the authors show that targeted degradation of one of its target mRNAs, PRMT1, a known negative regulator of autophagy, promotes autophagy.

Overall, the experiments presented in this paper are well thought out and well executed. However, there were a few aspects in the paper that could have been further explored.

➤ **We are grateful for the reviewer's time and thorough review of our work.**

1. On page 10, the authors ask, "when and where does LC3B bind to LMD substrates within the cells?" The authors show through fractionation experiments that LC3B-I and LC3B-II were primarily found in the cytoplasmic fraction. However, multiple studies have shown that LC3 cycles in and out of the nucleus. In 2015, it was shown by Huang, et al., (Mol Cell) that LC3 shuttles between the nucleus and the cytoplasm through the action of the SIRT1 deacetylase. I am curious to see how mutations in LC3B that prevent their shuttling from the nucleus to the cytoplasm affect substrate binding and subsequent LMD.

- **The reviewer has suggested interesting experiments. We found that a significant amount of LC3B was present in the nucleus and redistributed into the cytoplasm upon Rapa and CQ treatment (Fig. 5f in the revised manuscript). Therefore, our data were not inconsistent with those of previous reports.**
- **All gene regulatory processes, including transcription, splicing, mRNA export, and protein shuttling, might affect the cytoplasmic level of LC3B protein. In**

this study, we found that LMD largely occurs in the cytoplasm. Therefore, we expect that LC3B shuttling might influence LMD efficiency indirectly by regulating the cytoplasmic level of LC3B. Therefore, although the experiments suggested by the reviewer are very intriguing, they are outside the scope of this study.

2. Depletion of LC3B is not very efficient (Fig. S2a). Could a more efficient knockdown of LC3B affect the abundance of other identified cargo?

➤ **During western blotting, we always loaded 3-fold serial dilutions in the leftmost three or four lanes for the purpose of a standard. Using this standard, we compared the relative LC3B levels. We found that *LC3B* siRNA treatment caused a 3-fold reduction in the amount of LC3B under our conditions. We usually obtained a 3-5-fold reduction of a target protein using a specific siRNA. Therefore, the observed downregulation shown in Fig. S2a (Supplementary Fig. 3a in the revised manuscript) was sufficient for the transcriptome analysis in our study.**

3. It is interesting to see that the mRNA interaction with LC3B requires its ability to conjugate with PE (Fig. 3c), and mRNA, LC3B-II, and the CCR4-NOT1 complex are recruited to the phagophore (Fig. S9 and 10 and Fig. 4e) but mRNA is not degraded through autophagy. I think there could be more discussion about why the interaction between LC3B-II and mRNA is not enough to degrade the mRNA through autophagy (mRNA as a cargo) but needs the help of other proteins.

➤ **We appreciate the critical comments on LMD. As shown in this study, we showed that *PRMT1* mRNA degradation via LMD proceeds efficient autophagy. If LMD is inefficient, the downstream autophagy pathway is inhibited. Therefore, the following sentences have been added to the Discussion section: “Therefore, our observations suggest that, upon autophagic activation, LMD contributes to the creation of a suitable intracellular environment for efficient autophagy activity (degradation of cargo via formation of autolysosomes) by specifically degrading its target substrates before the formation of autolysosomes”.**

4. Is the LMD inhibited if there no interaction between LC3B and the CCR4-NOT1 complex? What is the mRNA or protein level of PRMT1 in the CNOT1 or CNOT7 knockdown or knockout cells? Also, what about the autophagy activity?

- **In our study, we found a specific association between LC3B and the CCR4-NOT complex within the cells. Although we did not determine whether this interaction occurs directly or indirectly, we observed that downregulation of a CCR4-NOT component inhibits LMD. Therefore, we concluded that efficient LMD depends on the CCR4-NOT complex.**
- **As suggested, we analyzed the relative levels of PRMT1 mRNA, PRMT1 protein, and p62 protein (for measuring autophagy activity) in cells depleted of CNOT1 or CNOT7. First, downregulation of either CNOT1 or CNOT7 increased the levels of endogenous PRMT1 mRNA following treatment with Rapa and CQ (Fig. 6e, f in the revised manuscript). Second, downregulation of either CNOT1 or CNOT7 increased the protein levels of endogenous PRMT1 and p62. These new data have been added to Supplementary Fig. 13d in the revised manuscript.**

5. Does the binding between LC3B and CNOT1 or CNOT7 need LC3B lipidation? Is the increasing binding after rapamycin treatment (Fig. 3d) due to the higher affinity to LC3B-II? Can these two proteins bind to LC3B that lacks the ability to conjugate with PE, such as LC3BG120A?

- **This comment is also related to comment #6 of reviewer #2. Reviewer #2 suggested employing cells depleted of ATG5 and ATG7. The new experiments revealed that treatment with Rapa and CQ increased the association between LC3B and CNOT1 (and CNOT7), which was reversed by downregulation of both ATG5 and ATG7. These data indicate that the association between LC3B and CNOT1 (and CNOT7) is promoted by LC3B conversion. These new data have been added to Fig. 6b and c in the revised manuscript. Please also refer to our responses to comment #6 of reviewer #2.**

6. Besides PRMT1, there are some other genes with a big change in mRNA level before and after knocking down LC3 as seen in Fig. S4, such as DDIT4, which has been shown

to regulate mTOR (Foltyn et al, 2019), so that it may have a role in autophagy regulation. Are there any other ways that LMD regulates autophagy?

- **We appreciate the reviewer’s insightful comments on our transcriptome data. As suggested, we added the following sentences to the Discussion section: “In this study, we propose that the rapid degradation of *PRMT1* mRNA via LMD facilitates the autophagy process. However, in addition to *PRMT1* mRNA, LMD may target a variety of cellular mRNAs involved in autophagy. For instance, DNA damage-inducible transcript 4 (DDIT4), which is known to repress mTORC1-mediated signaling, was identified as an LMD substrate in our transcriptomic analysis (Supplementary Fig. 5). A recent study showed that DDIT4 induction leads to an excessive release of reactive oxygen species, resulting in impaired autophagy in dry eye disease. Therefore, proper coordination of various LMD substrates before autolysosome formation might be a prerequisite for efficient autophagy.”**

7. The sections involving analyses with regards to CCR4-NOT interaction could be better written. For example, the analysis of experiments found in Extended Figure 9-10 and parts of Fig. 3 was written in the Discussion section. This could have been included on page 12 when Fig. 3 was initially discussed. A re-write could make this section of the paper flow better.

- **As commented, Figures were reorganized, and proper description was also added to the Results section in the revised manuscript.**

8. The introduction section also seems to be inadequate, especially when considering that Nature Communications has a broad reach.

- **The manuscript was initially submitted to another journal belonging to the Nature Publishing Group. After rejection, the manuscript was transferred to *Nature Communications*. As pointed out by the reviewer, we reorganized all Figures and added the Introduction section during revision, according to the journal format.**

Figure R1

* - possible false positive

Rank	Motif	P-value	log P-value	% of Targets	% of Background	STD(Bg STD)	Best Match/Details	Motif File
1		1e-224	-5.165e+02	15.97%	14.43%	4.6bp (4.7bp)	hsa-miR-340 MIMAT0004692 Homo sapiens miR-340 Targets (miRBase)(0.647) More Information Similar Motifs Found	motif file (matrix)
2		1e-87	-2.006e+02	12.98%	12.10%	4.7bp (4.9bp)	hsa-miR-3909 MIMAT0018183 Homo sapiens miR-3909 Targets (miRBase)(0.675) More Information Similar Motifs Found	motif file (matrix)
3		1e-76	-1.768e+02	5.30%	4.75%	4.9bp (4.9bp)	hsa-miR-4262 MIMAT0016894 Homo sapiens miR-4262 Targets (miRBase)(0.824) More Information Similar Motifs Found	motif file (matrix)
4		1e-70	-1.627e+02	5.99%	5.44%	4.9bp (5.0bp)	hsa-miR-4517 MIMAT0019054 Homo sapiens miR-4517 Targets (miRBase)(0.679) More Information Similar Motifs Found	motif file (matrix)
5		1e-57	-1.315e+02	3.67%	3.28%	4.6bp (4.8bp)	hsa-miR-4313 MIMAT0016865 Homo sapiens miR-4313 Targets (miRBase)(0.676) More Information Similar Motifs Found	motif file (matrix)
6		1e-41	-9.641e+01	2.65%	2.36%	4.6bp (5.0bp)	hsa-let-7a* MIMAT0004481 Homo sapiens let-7a* Targets (miRBase)(0.641) More Information Similar Motifs Found	motif file (matrix)
7		1e-38	-8.879e+01	4.57%	4.21%	4.8bp (5.1bp)	hsa-miR-874 MIMAT0004911 Homo sapiens miR-874 Targets (miRBase)(0.704) More Information Similar Motifs Found	motif file (matrix)
8		1e-37	-8.596e+01	2.98%	2.69%	4.7bp (4.8bp)	hsa-miR-4641 MIMAT0019701 Homo sapiens miR-4641 Targets (miRBase)(0.659) More Information Similar Motifs Found	motif file (matrix)
9		1e-36	-8.364e+01	1.95%	1.72%	4.8bp (5.3bp)	hsa-miR-602 MIMAT0003270 Homo sapiens miR-602 Targets (miRBase)(0.776) More Information Similar Motifs Found	motif file (matrix)
10		1e-36	-8.296e+01	2.17%	1.93%	4.7bp (4.6bp)	hsa-miR-9 MIMAT0000441 Homo sapiens miR-9 Targets (miRBase)(0.785) More Information Similar Motifs Found	motif file (matrix)
11		1e-35	-8.166e+01	2.01%	1.78%	4.7bp (4.8bp)	hsa-miR-1280 MIMAT0005946 Homo sapiens miR-1280 Targets (miRBase)(0.663) More Information Similar Motifs Found	motif file (matrix)
12		1e-30	-7.019e+01	6.31%	5.93%	4.7bp (5.2bp)	hsa-miR-4472 MIMAT0018999 Homo sapiens miR-4472 Targets (miRBase)(0.695) More Information Similar Motifs Found	motif file (matrix)
13		1e-29	-6.753e+01	1.47%	1.29%	4.8bp (5.3bp)	hsa-miR-486-3p MIMAT0004762 Homo sapiens miR-486-3p Targets (miRBase)(0.688) More Information Similar Motifs Found	motif file (matrix)
14		1e-28	-6.625e+01	2.13%	1.92%	4.8bp (4.8bp)	hsa-miR-3909 MIMAT0018183 Homo sapiens miR-3909 Targets (miRBase)(0.687) More Information Similar Motifs Found	motif file (matrix)
15		1e-26	-6.192e+01	0.36%	0.28%	4.7bp (4.1bp)	hsa-miR-3912 MIMAT0018186 Homo sapiens miR-3912 Targets (miRBase)(0.692) More Information Similar Motifs Found	motif file (matrix)
16		1e-24	-5.641e+01	3.53%	3.28%	4.7bp (4.7bp)	hsa-miR-522* MIMAT0005451 Homo sapiens miR-522* Targets (miRBase)(0.777) More Information Similar Motifs Found	motif file (matrix)
17		1e-16	-3.759e+01	0.33%	0.27%	4.7bp (5.1bp)	hsa-miR-4500 MIMAT0019036 Homo sapiens miR-4500 Targets (miRBase)(0.835) More Information Similar Motifs Found	motif file (matrix)

Figure R1. Motif analysis using sequences in the 3'UTR without peaks

Figure R2. CLIP-seq data after removing PCR duplicates using Picard (MarkDuplicates). a, metagene analysis after removing PCR duplicates. b, Motif analysis after removing PCR duplicates.

Figure R3. Consensus motif for LC3B binding. The consensus RNA sequences from LC3B peaks located in the 5'UTR, CDS, or 3' UTR in the cells treated with either DMSO (left) or Rapa + CQ (right) were predicted by MEME (upper) or HOMER (lower).

Figure R4

Figure R4. CDF plots for the relative changes in the half-life of mRNAs upon LC3B downregulation in control cells.

REVIEWERS' COMMENTS

Reviewer #1 (Remarks to the Author):

The authors have addressed most of my concerns. I believe this to be an exciting manuscript with important implications bridging two fields.

The issues that remain can be discussed and or qualified in the manuscript.
The main issue is establishing LC3B as an RBP as this seems to be a very novel part of the paper.

The derived affinity of LC3B against CLIP-derived motifs is rather weak ~50uM kd.
This is lower than most well-established RBPs. The authors should contextualize their findings based on what is known in the field about RNA-protein interactions. Could the mechanism of binding in cells require additional co-factors that account for the weak binding in vitro? The authors should mention the possibility that LC3B may in-part recognize RNA directly or semi-directly via interactions with the PolyA machinery.

Second, the mechanism of binding remains unclear. In response to my comments regarding R-rich motifs mediating the binding, the authors provide Refs 28-30 in the main text.
2 of those papers/reviews (28 and 30) discuss R and RG motifs commonly found in human intrinsically disordered protein domains (which appear to interact with structured RNA such as rG4s) while the other (29) focuses primarily on viral RBDs that commonly interact with structure RNAs. To be clear, it is completely possible that the RRR motif is important for binding, however the expressed motivation for selecting these aa is not well-justified.

In the reviewer response: "First of all, we would like to mention that many SR-rich motifs within RBPs are not disordered. They lie within a well-structured helix rather than a disordered region."
It would be greatly appreciated if the authors can provide the evidence/citation that the SR domains of RBPs are not disordered.

"To address the reviewer's comments, we carried out several experiments as follows: First, we performed in vivo CLIP experiments, which allowed us to monitor a direct interaction between RNA and protein within cells, using either LC3B-WT or R/Q. The in vivo CLIP results showed that LC3B-WT, but not the LC3B-R/Q mutant, directly binds to endogenous target mRNAs (PRMT1 mRNAs and MARS1 mRNAs). These new data have been added to Supplementary Fig. 6 in the revised manuscript."

Finally, it cannot be assumed that CLIP derives only direct interactions. The issue of background has been previously discussed in the a number of reviews and papers (for example Yeo Lab UCSD and Ule Lab Crick Inst.). CLIP has been applied to non-RBPs (Khavari Lab Stanford) and while non-RBPs crosslink more weakly, they still crosslink. CLIP has been used to study protein complexes (Black Lab UCLA). Thus it cannot be assumed that CLIP derives exclusively direct RBP interactions especially when the polyA signal is being retrieved. PolyA signal has numerous binding RBPs that interact with high affinity and is generally a core-motif.

With respect to figure s6, the difference in the western blot banding pattern and overall expression of proLC3B-R/Q (Fig S6) needs to be discussed. It appears that the mutant is expressed at lower levels and is primarily the shorter form even pre-treatment. Could this impact the interpretation of the CLIP-PCR.

[[[
SEP]

[[[
SEP]

[[[
SEP]

Reviewer #2 (Remarks to the Author):

The authors have done a great job addressing the comments resulting in a strengthened manuscript. I have no further comments to add. Congratulations on your very interesting discovery.

Reviewer #3 (Remarks to the Author):

I think the authors have answered all of my questions and I have no further comments.

** See Nature Research's author and referees' website at www.nature.com/authors for information about policies, services and author benefits

Responses to reviewer #1's comments

The authors have addressed most of my concerns. I believe this to be an exciting manuscript with important implications bridging two fields.

- **We appreciate the reviewer's interest in our study and the valuable comments. Please see our detailed responses below.**

The issues that remain can be discussed and or qualified in the manuscript.

The main issue is establishing LC3B as an RBP as this seems to be a very novel part of the paper.

The derived affinity of LC3B against CLIP-derived motifs is rather weak ~50uM kd. This is lower than most well-established RBPs. The authors should contextualize their findings based on what is known in the field about RNA-protein interactions. Could the mechanism of binding in cells require additional co-factors that account for the weak binding in vitro? The authors should mention the possibility that LC3B may in-part recognize RNA directly or semi-directly via interactions with the PolyA machinery.

- **We appreciate the reviewer's critical comments on our work and we agree with the reviewer. In the Discussion section of the previous submission, we had mentioned this possibility raised by the reviewer. During revision, we have added more description, emphasising the possible contribution of other RBPs or RNA structures to promoting LC3B binding to AAUAAA motif, as follows: "Under normal conditions, LC3B preferentially associates with target mRNAs harboring the AAUAAA motif (Fig. 1). Although general RBPs bind to their target RNAs with dissociation constant ranging from nanomolar to micromolar values {Kim, 2021 #97}, our fluorescence polarization assay showed a low RNA-binding affinity of purified recombinant LC3B to the AAUAAA motif (Fig. 1f). However, the association and specific loading of LC3B onto target mRNAs within cells is expected to be guided by RNA structures and other RBPs upstream of the AAUAAA motif (Fig. 4)."**

Second, the mechanism of binding remains unclear. In response to my comments

regarding R-rich motifs mediating the binding, the authors provide Refs 28-30 in the main text. 2 of those papers/reviews (28 and 30) discuss R and RG motifs commonly found in human intrinsically disordered protein domains (which appear to interact with structured RNA such as rG4s) while the other (29) focuses primarily on viral RBDs that commonly interact with structure RNAs. To be clear, it is completely possible that the RRR motif is important for binding, however the expressed motivation for selecting these aa is not well-justified.

- **When starting this work, we predicted the arginine-rich motif based on the literature. As we described in the first paragraph of the Results section of the previous submission, several studies have also suggested that the arginine-rich motif within LC3 is involved in the association between LC3B and RNAs. Therefore, to further justify our rationale, we have added the following sentence in the Results section of the revised manuscript: “Furthermore, the arginine-rich motif has been implicated to be involved in association between LC3B and fibronectin mRNA²³⁻²⁵”.**

In the reviewer response: “First of all, we would like to mention that many SR-rich motifs within RBPs are not disordered. They lie within a well-structured helix rather than a disordered region.” It would be greatly appreciated if the authors can provide the evidence/citation that the SR domains of RBPs are not disordered.

- **We apologize for this poor description. The above sentence was meant to be as follows: “Many SR-rich motifs within RBPs are disordered. However, once the SR-rich motifs associate with target RNAs, they can become structured.”**

“To address the reviewer’s comments, we carried out several experiments as follows: First, we performed in vivo CLIP experiments, which allowed us to monitor a direct interaction between RNA and protein within cells, using either LC3B-WT or R/Q. The in vivo CLIP results showed that LC3B-WT, but not the LC3B-R/Q mutant, directly binds to endogenous target mRNAs (PRMT1 mRNAs and MARS1 mRNAs). These new data have been added to Supplementary Fig. 6 in the revised manuscript.”

Finally, it cannot be assumed that CLIP derives only direct interactions. The issue of

background has been previously discussed in the a number of reviews and papers (for example Yeo Lab UCSD and Ule Lab Crick Inst.). CLIP has been applied to non-RBPs (Khavari Lab Standford) and while non-RBPs crosslink more weakly, they still crosslink. CLIP has been used to study protein complexes (Black Lab UCLA). Thus it cannot be assumed that CLIP derives exclusively direct RBP interactions especially when the polyA signal is being retrieved. PolyA signal has numerous binding RBPs that interact with high affinity and is generally a core-motif.

- **We agree with the reviewer’s comment. We have revised the description of CLIP-seq as follows: “This approach mainly allows us to monitor the direct interaction between RNA and protein within cells”.**

With respect to figure s6, the difference in the western blot banding pattern and overall expression of proLC3B-R/Q (Fig S6) needs to be discussed. It appears that the mutant is expressed at lower levels and is primarily the shorter form even pre-treatment. Could this impact the interpretation of the CLIP-PCR.

- **We agree with reviewer’s comments, and we were also curious about this phenomenon. A possible interpretation has been added to the Results section of the revised manuscript as follows: “It should be noted that we observed a differential protein migration pattern upon SDS-PAGE when samples of wild-type and R/Q mutant of proLC3B were used. A previous study reported the same phenomenon²⁵. In addition, purified recombinant LC3B-R/Q migrated faster migration than LC3B-WT (Supplementary Fig. 2a). Currently, we do not know the exact reasons for this difference. Probably, a change in protein charge after introducing substitution may affect the migration pattern of proLC3B and efficiency of proLC3B conversion.”**
- **In the case of Supplementary Figure 6d, we did not directly compare the levels of co-IPed mRNA in IP of LC3b-WT with those in IP of LC3B-R/Q. Instead, we normalized the levels of co-IPed mRNA under Rapa- and CQ-treated conditions to those of co-IPed mRNA under untreated conditions. In other words, we assessed a difference in RNA-binding efficiency of each protein under untreated conditions and treated conditions. In this way, we minimized a possible effect due to differences in protein expression and IP efficiency.**